# COMPOSITION OF PRETRAINED DIFFUSION MODELS: A LOGIC-BASED CALCULUS

**Peter Blohm**
Aalto University
`peter.blohm@aalto.fi`

**Vikas Garg**
Aalto University and YaiYai Ltd
`vgarg@csail.mit.edu`

## ABSTRACT

Composing pretrained diffusion models provides a cost-effective mechanism for encoding constraints and unlocking complex generative capabilities. Prior work relies on crafting compositional operators that seek to extend set-theoretic notions such as union and intersection to diffusion models, e.g., using a product or mixture of the underlying energy functions. We expose the inadequacy and inconsistency of combining these operators, including limited mode coverage, biased sampling, instability under negation queries, and failure to satisfy basic compositional laws such as idempotency and distributivity. We introduce a principled calculus grounded in fuzzy logic that resolves these issues. Specifically, we define a general class of conjunction, disjunction, and negation operators that generalize the classical mixture-, product-, and harmonic-mean-style operators, illustrating how they circumvent various pathologies and enable precise combinatorial reasoning with score models. Beyond existing methods, the proposed *Dombi* operators yield complex generative outcomes, such as XOR-style logical compositions of pretrained score models. We establish rigorous theoretical guarantees on the stability of Dombi compositions, and derive Feynman-Kac correctors to mitigate the sampling bias in score composition. Empirical results on image generation with Stable Diffusion and multi-objective molecular generation substantiate the conceptual, theoretical, and methodological benefits. Overall, this work lays the foundation for systematic design, analysis, and deployment of diffusion ensembles. Code is available at `github.com/Aalto-QuML/logic-diffusion-composition`.

## 1 INTRODUCTION

Pretrained general-purpose generative machine learning models (Devlin et al., 2019; Brown et al., 2020) have become practically synonymous with the term artificial intelligence itself. Their vast capabilities (Bommasani, 2021; Wei et al., 2022), however, come at the cost of an excessive need for growing datasets (Kaplan et al., 2020; Villalobos et al., 2022), and yet additional techniques are needed to reach adequate performance in downstream tasks. Finetuning (Devlin et al., 2019), human-feedback-based reinforcement learning (Christiano et al., 2017; Ouyang et al., 2022; Zhang et al., 2023), retrieval augmented generation (Lewis et al., 2020), or even specialized prompting techniques (Brown et al., 2020) are then used to retrofit models to specialized tasks and domains.

As an alternative to monolithic general models, compositional generation (Jordan & Jacobs, 1994; Hinton, 1999; 2002; Yuksel et al., 2012; Vedantam et al., 2018; Du et al., 2020) seeks to combine the domain knowledge from different models—or different guidance signals—to solve a task at hand. As many models follow probabilistic formulations, using probabilistic language for composition is a natural approach. Products of Experts (PoEs) (Hinton, 1999; 2002; Liu et al., 2022; Du et al., 2023; Skreta et al., 2025a) have been devised and widely used as a mechanism to enforce conjunctive constraints, with the idea that their product is only large when all components are large. The assumption underlying this approach to model joint distributions—statistical independence of the factors—however, does not in general hold.

Often tackled as a separate problem is the concept *avoidance* in generation. Similar to other tasks, *unlearning* (Ginart et al., 2019; Nguyen et al., 2022; Wang et al., 2024) as a specific form of fine-

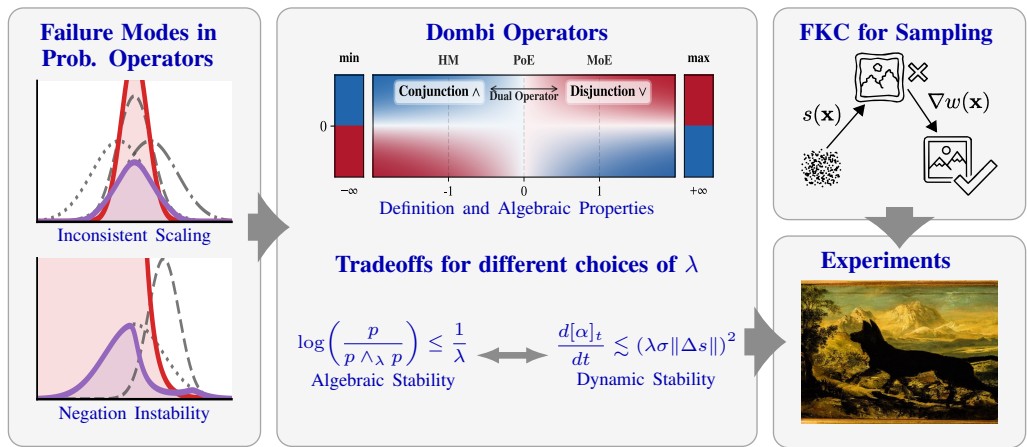

Figure 1: Overview of Paper Structure and Main Contributions.

tuning or post-training *avoidance* and steering methods (Dhariwal & Nichol, 2021; Ho & Salimans, 2021; Dong et al., 2023; Garipov et al., 2023; Kirchhof et al., 2025) have been proposed, which often utilize PoE with inverse probability densities for avoidance or rely on training additional models.

In this paper, we investigate the composition of diffusion models from a fuzzy set-theoretic and fuzzy-logic perspective. We propose a procedure to derive sets of well-behaved composition operators, and among them, propose *Dombi operators* in Section 4 as a one-parameter family, extending and uniting commonly used operations such as mixture of experts (Jordan & Jacobs, 1994) (MoE), harmonic mean (Garipov et al., 2023) (HM), and as a special case, the geometric mean—a tempered Product of Experts (Hinton, 1999) (PoE), as visualized in Figure 3. In contrast to many existing effective methods, our approach is purely online and utilizes pretrained diffusion models and online density estimates. An overview of our main contributions is provided in Figures 1 and 6.

## 2 BACKGROUND AND RELATED WORK

### 2.1 SCORE-BASED MODELS

We want to approximate a probability distribution $p$ defined over $\mathbb{R}^d$ to sample from it. In the context of score-based modeling (Song et al., 2021), we first recast $p$ as a Boltzmann distribution, and let the model learn the *score function* $s_\theta(\mathbf{x}) \approx \nabla \log p(\mathbf{x})$, avoiding the unknown partition function. To facilitate sampling via MCMC, the data distribution $p$ is gradually destroyed according to the forward noising SDE (Øksendal, 2003)

$$d\mathbf{x}_\tau = f_\tau(\mathbf{x})d\tau + \sigma_\tau d\overline{\mathbf{w}}_\tau, \qquad \mathbf{x}_0 \sim p(\mathbf{x}_0).$$

Here $f_\tau : \mathbb{R}^d \to \mathbb{R}^d$ is some, usually linear, drift function and $\sigma_\tau : \mathbb{R} \to \mathbb{R}$ is a time-dependent diffusion coefficient and $\overline{\mathbf{w}}_\tau$ is the Wiener process. These functions are chosen such that $\mathbf{x}_{\tau=1} \sim \mathcal{N}(0, \mathbf{I}_d)$, the standard Gaussian. For sampling, we simulate the backward process with $t = 1 - \tau$ as

$$d\mathbf{x}_t = \left[-f_t(\mathbf{x}_t) + \sigma_t^2 \nabla_\mathbf{x} \log p_t(\mathbf{x}_t)\right] dt + \sigma_t d\mathbf{w}_t. \tag{1}$$

which satisfies the Fokker-Planck equation

$$\frac{\partial p_t(\mathbf{x})}{\partial t} = -\langle\nabla, p_t(\mathbf{x})(-f_t + \sigma_t^2 \nabla_\mathbf{x} \log p_t(\mathbf{x}))\rangle + \frac{\sigma_t^2}{2}\Delta p_t(\mathbf{x}), \tag{2}$$

where $\Delta p_t$ denotes the Laplacian of $p_t$ and $\langle\nabla, \cdot\rangle$ is the divergence operator. For the rest of this paper, we assume we are given a set of pretrained score models $\{s_t^i\}_{i=1}^k$, which model the respective probability distributions $\{p_t^i\}_{i=1}^k$. For statements about the $t = 1$, we omit the index.

To translate the theory developed in this paper to practice, we rely on efficient density estimation to assign responsibility to score functions. We can efficiently estimate densities during inference with Itô's Lemma (Karczewski et al., 2025a; Skreta et al., 2025b) as

$$d\log p_t(\mathbf{x}_t) \approx \langle d\mathbf{x}_t, \ s_t(\mathbf{x}_t)\rangle + \left(\langle\nabla, \ f_t(\mathbf{x}_t)\rangle + \langle f_t(\mathbf{x}_t), \ s_t(\mathbf{x}_t)\rangle - \frac{\sigma_t^2}{2}\|s_t(\mathbf{x}_t)\|^2\right)dt. \tag{3}$$

## 2.2 Composition of Score Fields

There is a quickly growing body of work on compositions, mixtures, and products of energy-based models (EBMs), as well as flow and diffusion models. We explicitly focus on *training-free* mixtures of score functions in diffusion. Prior work (Du et al., 2020; Ho & Salimans, 2021; Skreta et al., 2025a;b; Gaudi et al., 2025) mainly bases composition on probabilistic operations on the underlying distributions. As the interpretation of these operations is often logical or set-theoretic, we will use the symbols $\{\vee, \wedge, \neg\}$ to denote them, for both probability densities and their scores. In score-based modelling, conjunctions are then usually represented by (sometimes geometric) products

$$p^1(\mathbf{x}) \wedge_\times p^2(\mathbf{x}) \coloneqq p^1(\mathbf{x})p^2(\mathbf{x}) \implies s^1(\mathbf{x}) \wedge_\times s^2(\mathbf{x}) = s^1(\mathbf{x}) + s^2(\mathbf{x}) \tag{4}$$

and disjunctions by mixtures, where we use the weighting $\alpha^i = \frac{p^i(\mathbf{x})}{p^1(\mathbf{x})+p^2(\mathbf{x})}$, with

$$p^1(\mathbf{x}) \vee_+ p^2(\mathbf{x}) \coloneqq \frac{1}{2}p^1(\mathbf{x}) + \frac{1}{2}p^2(\mathbf{x}) \implies s^1(\mathbf{x}) \vee_+ s^2(\mathbf{x}) = \alpha^1 s^1(\mathbf{x}) + \alpha^2 s^2(\mathbf{x}). \tag{5}$$

Two noteworthy exceptions from product-based conjunctions are Garipov et al. (2023), who model conjunctions with the *harmonic mean* $p^1(\mathbf{x})p^2(\mathbf{x})/\left(p^1(\mathbf{x}) + p^2(\mathbf{x})\right)$ and Skreta et al. (2025b), who reweigh individual scores to steer towards equal density directly.

Importantly, under the usual dynamics of diffusion processes, for $t \neq 1$, nonlinear compositions do not commute with the noising operator, i.e., $p_t^1 \vee_+ p_t^2 = (p^1 \vee_+ p^2)_t$ but $p_t^1 \wedge_\times p_t^2 \neq (p^1 \wedge_\times p^2)_t$. This means that naive composition of perturbed score models leads to a bias that can be corrected with methods like sequential Monte Carlo (SMC) (Skreta et al., 2025a; Thornton et al., 2025). The typical formulation of Equations (1) and (2) is then extended to *weighted* SDEs, where samples have time-dependent log-weights $w_t$ which are defined via the weight field $g_t(\mathbf{x})$ as

$$dw_t = \overline{g}(\mathbf{x}_t)dt \implies \frac{\partial p_t(\mathbf{x})}{\partial t} = \overline{g}_t(\mathbf{x})p_t(\mathbf{x}), \quad \text{with} \quad \overline{g}(\mathbf{x}) \coloneqq g_t(\mathbf{x}) - \int g_t(\mathbf{x})p_t(\mathbf{x})d\mathbf{x}.$$

These weighted SDEs with $g_t(\mathbf{x})$ then must satisfy the Feynman-Kac PDE

$$\frac{\partial p_t(\mathbf{x})}{\partial t} = -\langle \nabla, p_t(\mathbf{x})(-f_t + \sigma_t^2 \nabla_\mathbf{x} \log p_t(\mathbf{x}))\rangle + \frac{\sigma_t^2}{2}\Delta p_t(\mathbf{x}) + \overline{g}_t(\mathbf{x})p_t(\mathbf{x}). \tag{6}$$

For nonlinear score operations like annealing, CFG, or PoE, Skreta et al. (2025a) then explicitly derive the biases incurred by approximating the true composed distribution with the composition of noisy scores, collect the "left-over" terms in $g$, and use additional correction methods. We adapt their formalism to improve the simulation of our operators in Section 4.

To *avoid* certain distributions, EBM's and score models are usually only negated *relative* to others (Vedantam et al., 2018; Du et al., 2020; 2023; Garipov et al., 2023; Dong et al., 2023; Skreta et al., 2025a; Gaudi et al., 2025), as also done in classifier-free guidance (Ho & Salimans, 2021) (CFG). In these settings, independent concept negation (ICN) for a concept $y$ is often defined, for $0 < \gamma < 1$ as $p(\mathbf{x}|\neg y) \propto p(\mathbf{x})/p(\mathbf{x}|y)^\gamma$ in the EBM context (Hinton, 2002; Du & Kaelbling, 2024). In more recent work (Liu et al., 2022; Du et al., 2020; Ho & Salimans, 2021), often the formulation $p(\mathbf{x}|\neg y) \propto p(\mathbf{x})^{1+\gamma}p(\mathbf{x}|y)^{-\gamma}$ used instead, derived via Bayes rule.

From a perspective of logic, these variants make use of the reciprocal as pseudo-inverse $\neg p(\mathbf{x}|y) = 1/p(\mathbf{x}|y)$, but to our knowledge, explicit negations in score-models are not often explored or theoretically justified, and alternatives (Chang et al., 2024) also lack clear theoretical interpretation.

## 2.3 Fuzzy Logic

Our proposed method directly draws from the theory of fuzzy logic. Fuzzy logic relaxes classical logic from a binary domain to real-valued *memberships* in $[0, 1]$. We follow the definitions and notation from Klement et al. (2013) for the following concepts. We define a *t-norm*, a generalization of conjunction or intersection operations, as $T : [0, 1]^2 \to [0, 1]$ which is commutative, associative, monotonously increasing in either variable, and fulfills the boundary condition $\forall x \in [0, 1] : T(x, 1) = x$. With the standard negation $N(x) = 1 - x$, we define the *dual t-conorm* $S : [0, 1]^2 \to [0, 1]$ of $T$, a disjunction, via De Morgan's law as $S(x, y) = N(T(N(x), N(y)))$.

T-norms that are *strict*, i.e., continuous and strictly increasing, can be *generated* (Dombi, 1982; Klement et al., 2013) by a continuous, strictly decreasing function $f : [0, 1] \to [0, \infty]$ with $f(1) = 0$,

as so-called *additive generator*, i.e., $T(x, y) \coloneqq f^{-1}(f(x) + f(y))$. For this work, the parametrised Dombi t-norm is the most important representative, generated by $f_\lambda(x) = (\frac{1}{x} - 1)^\lambda$. A favorable property of the Dombi t-norm is that $\lim_{\lambda \to \infty} T_\lambda = T_M = \min$. The min t-norm $T_M$ together with $S_M = \max$ is the *only* continuous De Morgan dual that is idempotent with $T_M(x, x) = x$ and distributive with $T_M(x, S_M(y, z)) = S_M(T_M(x, y), T_M(x, z))$ (Klement et al., 2013). To make the domain of probability densities compatible with the theory of fuzzy logic, we utilize some bijective, order-preserving function $\phi : \mathbb{R}_{\geq 0} \cup \{\infty\} \to [0, 1]$ which converts densities into fuzzy membership.

## 3 FAILURE MODES IN SCORE COMPOSITION

We provide further motivation for our approach with a brief illustration of the mismatch between expectations and actual behaviour in score composition when using PoE and MoE as operators. We illustrate clear stability concerns when avoiding concepts, as well as unintended biases in the sampling dynamic caused by score rescaling. Finally, we demonstrate that interpreting PoE and MoE as set operators easily leads to modelling mistakes.

### 3.1 NEGATION INSTABILITY

The two distinct EBM-style negation $p^1(\mathbf{x})/p^2(\mathbf{x})^\gamma$ and CFG-style negation $p^1(\mathbf{x})^{1+\gamma}/p^2(\mathbf{x})^\gamma$ operators both easily translate to score calculus. However, specifically, the EBM-style negation shifts the target distribution and requires careful calibration of $\gamma$ to avoid exploding tails (Garipov et al., 2023; Chang et al., 2024; Ban et al., 2024). The commonly used CFG-style negation behaves much more stably in practice, yet theoretical arguments for its use remain limited. While better behaved, CFG negation might *overaccentuate* regions in the target distribution $p^1$ where $p^2$ vanishes (Chidambaram et al., 2024). In conclusion, neither version of this operator guarantees a normalizable distribution and both tend to exhibit unwanted tail behaviour as depicted in Figure 2a. We revisit this issue in Section 4.2.

### 3.2 INCONSISTENT TEMPERATURE SCALING

PoE uses the additive score calculus $s^1 \wedge_\times s^2 \coloneqq s^1 + s^2$. This leads to a scaling of scores depending on their alignment: $\|s^1 \wedge_\times s^2\| = \sqrt{\|s^1\|^2 + \|s^2\|^2 + 2\|s^1\|\|s^2\| \cos\theta}$, where $\theta$ is the angle between $s^1, s^2$. In diffusion, temperature scaling—scaling of the score vectors—is one of the main methods to control the behavior of the model (Guo et al., 2017; Chidambaram et al., 2024; Karczewski et al., 2025a;b). Crucially, the score alignment ($\theta$) may vary significantly depending on location and the models used.

In regions with high score alignment (small $\theta$), temperature is decreased, and the composition is biased towards higher density regions than what is dictated by any component. Conversely, in regions with low alignment between scores (large $\theta$), the temperature is increased, *discouraging* higher density regions. Figure 2b illustrates this behavior in contrast to the Dombi operators, which guarantee $\|s^1 \wedge_\lambda s^2\| \leq \max\{\|s^1\|, \|s^2\|\}$. This problem persists when scaling with, e.g., the geometric mean or *score averaging* $s^1(\mathbf{x})/2 + s^2(\mathbf{x})/2$, where the score norm still varies across positions.

### 3.3 COMPOSITION PROPERTIES

Model composition is often interpreted as a *logical/set* operation over the hypothetical sets of "typical" samples of each distribution. This interpretation leads to pitfalls with probabilistic operators, as MoE and PoE do not exhibit the necessary algebraic properties expected of logical or set operators. A simple example of this is avoiding *multiple* distributions $p^2$ and $p^3$ *individually*. Intuitively, one might use a conjunction over multiple negated distributions. The resulting operation, however, does not match the expected result, as negations and products commute:

$$p^1 \wedge_\times \neg p^2 \wedge_\times \neg p^3 = \boxed{\frac{p^1}{p^2 p^3} = p^1 \wedge_\times \neg(p^2 \wedge_\times p^3)} \neq \boxed{p^1 \wedge_\times \neg(p^2 \vee_+ p^3) = \frac{p^1}{p^2 + p^3}}.$$

In a more general sense, PoE is also neither idempotent, as $x \wedge_\times x = x^2 \neq x$ and distributes only in one direction, i.e., $(p^1 \wedge_\times p^2) \vee_+ p^3 \neq (p^1 \vee_+ p^3) \wedge_\times (p^2 \vee_+ p^3)$. This severely restricts the options for rewriting compositions for different purposes, such as collecting terms.

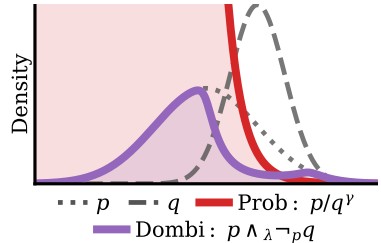 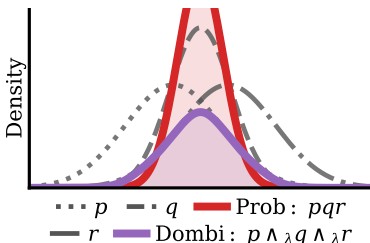

(a) Avoiding $q$: density ratios vs. Dombi.

(b) Intersection: products vs. Dombi.

Figure 2: Failure Modes of Composition with probabilistic approaches. (a) illustrates that density ratios can lead to unstable behavior, compared to the *bounded* referenced Dombi negation. (b) shows an intersection, where the product can lead to locally underscaled *and* overscaled temperatures, in contrast to Dombi composition.

## 4 DOMBI OPERATORS

In this section, we present our main contribution: a principled score-calculus based on the *Dombi t-norm* (Dombi, 1982). We present a more general definition and derivation of a class of De Morgan dual operators for scores and densities in Appendix B.

We extend the Dombi t-norm, generated by $f_\lambda(x) = \left(\frac{1}{x} - 1\right)^\lambda$, for $\lambda \in \mathbb{R}_{>0}$, and lift it from fuzzy memberships to densities in a semantically meaningful way. This is achieved by the *referenced mapping function* $\phi_c$ that expresses negation with *reference* to some (pointwise) constant function $c(\mathbf{x})$. This constant can be interpreted as a normalizing factor and serves as a fixed point for negations, but does not affect other operations. In the context of distributions, this normalization by a reference distribution $c(\mathbf{x})$ is analogous to the probability ratios used in CFG, or the PoE conjunction, e.g., presented by Liu et al. (2022). With abuse of notation, we will write $\phi_c(p(\mathbf{x})) := \phi_c(p; \mathbf{x}) = \frac{p(\mathbf{x})}{p(\mathbf{x}) + c(\mathbf{x})}$.

**Definition 4.1** (Dombi Operators). *Choose* $\lambda \in \mathbb{R}_{>0}$ *and a continuously differentiable function* $c : \mathbb{R}^d \to \mathbb{R}_{>0}$ *with* $s_c = \nabla_{\mathbf{x}} \log c$. *For* $f_\lambda(x) = \left(\frac{1}{x} - 1\right)^\lambda$ *and* $\phi_c(p(\mathbf{x})) = \frac{p(\mathbf{x})}{p(\mathbf{x}) + c(\mathbf{x})}$, *let* $\alpha_\lambda^i = \frac{\exp(\lambda \log p^i(\mathbf{x}))}{\sum_{j \in \{1,2\}} \exp(\lambda \log p^j(\mathbf{x}))}$. *The* Dombi operators *are the De Morgan dual operators induced by* $f_\lambda, \phi_c$:

$$\neg_c p(\mathbf{x}) := \frac{c(\mathbf{x})^2}{p(\mathbf{x})} \implies \neg_c s(\mathbf{x}) = 2s_c(\mathbf{x}) - s(\mathbf{x}) \tag{7}$$

$$p^1(\mathbf{x}) \wedge_\lambda p^2(\mathbf{x}) := \frac{p^1(\mathbf{x})p^2(\mathbf{x})}{\left(p^1(\mathbf{x})^\lambda + p^2(\mathbf{x})^\lambda\right)^{1/\lambda}} \implies s^1(\mathbf{x}) \wedge_\lambda s^2(\mathbf{x}) = \alpha_{-\lambda}^1 s^1(\mathbf{x}) + \alpha_{-\lambda}^2 s^2(\mathbf{x}) \tag{8}$$

$$p^1(\mathbf{x}) \vee_\lambda p^2(\mathbf{x}) := \left(p^1(\mathbf{x})^\lambda + p^2(\mathbf{x})^\lambda\right)^{1/\lambda} \implies s^1(\mathbf{x}) \vee_\lambda s^2(\mathbf{x}) = \alpha_\lambda^1 s^1(\mathbf{x}) + \alpha_\lambda^2 s^2(\mathbf{x}) \tag{9}$$

*A detailed derivation of this result can be found in Appendix B.3.*

This definition bears multiple remarkable properties. As De Morgan dual operators, Dombi operators have favorable algebraic properties, described in Proposition B.2. Most notably, Dombi operators approximate a *Lattice*, meaning they (approximately) adhere to the axioms of set operations. In addition to their algebraic properties, we can exactly recover established score operators.

### 4.1 RELATION OF DOMBI OPERATORS TO PROBABILISTIC DENSITY OPERATIONS

Dombi compositions over distributions are power norms, and with different choices for the exponent $\lambda$, we recover well-known operators (illustrated in Figure 3), such as min/max for $\lambda \to \infty$, a linear mixture/harmonic mean for $\lambda = 1$. For $\lambda \to 0$, Dombi composition is undefined on densities, yet the score calculus for $\lambda \to 0$ is equivalent to the geometric mean. Most notably, this parametrization shows which operators work well as a calculus, as visualized in Figure 3. The Dombi score calculus

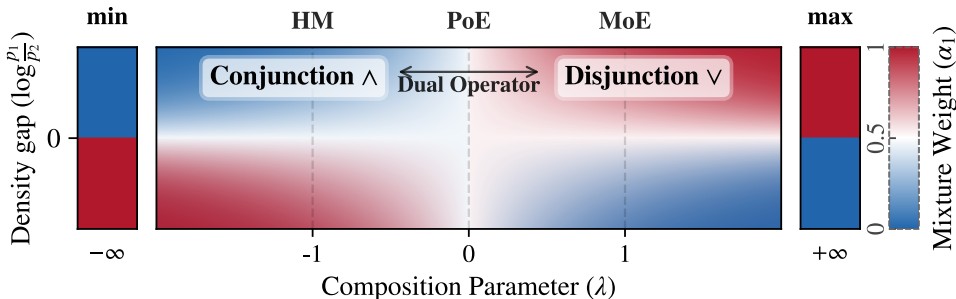

Figure 3: Visualisation of Dombi Composition $p_1(\mathbf{x}) \circ_\lambda p_2(\mathbf{x})$, depending on their density gap. Higher values of $\lambda$ result in more sensitive weights $\alpha$, and flipping the sign of $\lambda$ gives the De Morgan dual operator. Different choices of $\lambda$ correspond to known operators.

is also equivalent to a calculus induced by *power means* (Amari, 2007), which differ from the Dombi density operators by a constant factor of $2^{-1/\lambda}$. Power means are minimizers of corresponding $\alpha$-divergences—such as the KL-divergence—between sets of distributions. This connection suggests further parallels of score calculus to work on logical operators via divergences (Khalafi et al., 2025).

## 4.2 PROPERTIES OF REFERENCED NEGATION

Under our definition, referenced negation yields an expression equivalent to CFG-style negation for $\gamma = 1$. A crucial distinction between the Dombi calculus and vanilla CFG arises when negation is combined with conjunction. We know that a negated distribution is normalizable if the $\chi^2$ divergence is bounded. We then have, per definition (Nishiyama & Sason, 2020)

$$\chi^2(p||q) := \int \frac{(p(\mathbf{x}) - q(\mathbf{x}))^2}{q(\mathbf{x})} dx = \int \frac{p(\mathbf{x})^2}{q(\mathbf{x})} - 1 < \infty. \tag{10}$$

In practice, bounded $\chi^2$-divergence cannot be easily verified, but we can *guardrail* against diverging distributions with Dombi conjunctions. Dombi conjunctions are upper bounded by $\min$, so a single stable distribution prevents diverging composition results, regardless of the tail behaviour of other components. This is illustrated in Figure 2a, and justifies the use of unrestricted CFG as $\neg_{c,\gamma} p(\mathbf{x}) := c(\mathbf{x})^{1+\gamma}/p(\mathbf{x})^\gamma$. This non-involutive negation variant is not involutive, but allows adjusting negative guidance behavior —without the risk of divergence.

## 5 INFLUENCE OF $\lambda$ ON DISTRIBUTIVITY AND MIXTURE STABILITY

Besides the connection to prior work, the parameter $\lambda$ from the Dombi operators naturally appears as inverse temperature in the score composition. For $\lambda \to \infty$, the Dombi operators recover the exact $\{\min, \max\}$ lattice and with it distributive and idempotent behavior. For finite $\lambda$, the simple bounds in Proposition B.5 can be used to quantify biases in density compositions. We use this to present a simple bound for the maximal density bias we introduce when applying distributive laws.

**Corollary 5.1** (Idempotency and Distributivity Bias). *Let $\wedge_\lambda, \vee_\lambda$ be the Dombi density operators. From Proposition B.5 it follows that, for $B \in \mathbb{R}_{>0} : B \cdot 2^{\pm 2/\lambda} := [2^{-2/\lambda} B, 2^{2/\lambda} B]$*

$$\forall x \in \mathbb{R}_{\geq 0} : \quad x \vee_\lambda x = 2^{1/\lambda} x, \quad x \wedge_\lambda x = 2^{-1/\lambda} x \tag{11}$$

$$\forall x, y, z \in \mathbb{R}_{\geq 0} : \quad x \vee_\lambda (y \wedge_\lambda z) \in ((x \vee_\lambda y) \wedge_\lambda (x \vee_\lambda z)) \cdot 2^{\pm 2/\lambda}$$

$$\forall x, y, z \in \mathbb{R}_{\geq 0} : \quad x \wedge_\lambda (y \vee_\lambda z) \in ((x \wedge_\lambda y) \vee_\lambda (x \wedge_\lambda z)) \cdot 2^{\pm 2/\lambda} \tag{12}$$

These easily obtainable bounds trivially generalize to sharp bounds on arbitrary compositions, allowing us to make immediate statements about the stability of our composition. As our score coefficients vary during the inference process, we would naturally be interested in the rate of change of these coefficients, as drastic change rates might cause the composite model to "oscillate" between two scores, especially in conjunctions. As before, the statement can be extended to more complex formulas.

**Proposition 5.2** (Mixture Stability)**.** *Let* $\alpha_t = softmax_1(\lambda \log p_t^1, \lambda \log p_t^2)$, *for a Dombi composition* $p^1 \circ_\lambda p^2$. *Then it holds for the score difference* $\Delta s = s_t^1 - s_t^2$

$$|\mathbb{E}[d\alpha_t \mid \mathbf{x}_t]| = \mathcal{O}(|\lambda^2 - \lambda|\sigma_t^2\|\Delta s\|^2 dt), \qquad d[\alpha]_t = \mathcal{O}(\lambda^2\sigma_t^2\|\Delta s\|^2 dt)$$

Together, Corollary 5.1 and Proposition 5.2 quantify the tradeoff between compositional precision and mixture stability. High $\lambda$ results in small biases over the ground truth of the composition, but for large differences between the component scores $\|s_t^1 - s_t^2\|$, the mixing coefficients $\alpha^i$ might drastically oscillate. When $\lambda$ is chosen smaller, the volatility of the mixture is naturally bounded.

# 6 PRECISE SAMPLING WITH FEYNMAN-KAC CORRECTION

While Definition 4.1 explicitly states how the densities and consequently the scores of our target distribution look, simulation with, e.g., $d\mathbf{x}_t = \left[-f_t(\mathbf{x}_t) + \sigma_t^2(s_1(\mathbf{x}) \wedge_\lambda s_2(\mathbf{x}))\right] dt + \sigma_t d\overline{\mathbf{w}}$ will *not* sample from the desired marginals during the reverse process and consequently not from the correct target distribution $p_1(\mathbf{x}) \wedge_\lambda p_2(\mathbf{x})$. Skreta et al. (2025a) introduce *Feynman-Kac Correctors* (FKCs) for diffusion, which correct for the biases of score composition. We recast the composition with Dombi operators as weighted SDEs, then collect all terms that are missing from our score proposal into the weight field $g$. At inference time, SMC methods like systematic sampling can be used to correct for these biases.

In this section, we extend the FKC terms to our Dombi operators, and refer to Appendix C.1 for proofs. As the Dombi-composition just reduces to "power norms" of our densities, as well as a special case of geometric averages in the case of referenced negation, we present these two correction terms here. More complex compositions then propagate the weight-fields $g_t(\mathbf{x})$ of components.

**Proposition 6.1** (Referenced Negation as CFG+FKC, Skreta et al., 2025a)**.** *Consider two diffusion models* $q_t^1(\mathbf{x}), q_t^2(\mathbf{x})$ *defined via the Feynman-Kac equation in Equation* (6) *with weights* $g_t^i$. *The weighted SDE corresponding to the referenced negation of* $p_t \propto \neg_{q_t^2} q_t^1$ *is, with* $dw_t(\mathbf{x}) = g_t(\mathbf{x})dt$

$$
\begin{aligned}
d\mathbf{x}_t &= \left[-f_t(\mathbf{x}_t) + \sigma_t^2(2\nabla \log q_t^2(\mathbf{x}_t) - \nabla \log q_t^1(\mathbf{x}_t))\right] dt + \sigma_t d\overline{\mathbf{w}}_t \\
g_t(\mathbf{x}) &= \sigma_t^2\|\nabla \log q_t^1(\mathbf{x}_t) - \nabla \log q_t^2(\mathbf{x}_t)\|^2 + 2g_t^2(\mathbf{x}) - g_t^1(\mathbf{x}),
\end{aligned}
\tag{13}
$$

As stated in Equation (10), $p_t(\mathbf{x})$ is then a normalizable probability distribution, if and only if $\chi^2(q_t^2\|q_t^1) < \infty$. We might also want to anneal $q^2$ to tune the "narrowness" of the concept we avoid. We propose a combined annealing of the form $q^2(\mathbf{x})^{1+\gamma}/q^1(\mathbf{x})^\gamma$ to allow tuning the two distributions in relation to each other, while still maintaining slightly improved normalizability compared to the standard CFG, and maintaining an unbiased energy estimate for further composition.

Next, we state how FKC terms propagate through connectives. As both our connectives are essentially power-norms with positive or negative exponent, both cases can be handled at once.

**Theorem 6.2.** *Consider two weighted diffusion models* $q_t^1(\mathbf{x}), q_t^2(\mathbf{x})$ *defined via the Feynman-Kac equation with weights* $g_t^1(\mathbf{x}), g_t^2(\mathbf{x})$, *and a parameter* $\lambda \in \mathbb{R} \setminus \{0\}$. *The weighted SDE corresponding to* $p_t(\mathbf{x}) \propto \left(q_t^1(\mathbf{x})^\lambda + q_t^2(\mathbf{x})^\lambda\right)^{1/\lambda}$, *with* $\alpha_t^i = \frac{q_t^i(\mathbf{x})^\lambda}{q_t^1(\mathbf{x})^\lambda + q_t^2(\mathbf{x})^\lambda} \in (0,1)$, *and* $dw_t = \overline{g}_t(\mathbf{x})dt$ *is*

$$d\mathbf{x}_t = \left[-f_t(\mathbf{x}_t) + \sigma_t^2(\alpha_t^1 \nabla \log q_t^1(\mathbf{x}_t) + \alpha_t^2 \nabla \log q_t^2(\mathbf{x}_t))\right] dt + \sigma_t d\overline{\mathbf{w}}_t$$

$$g_t(\mathbf{x}) = (1-\lambda)\frac{\sigma_t^2}{2}\left[\left\|\sum_{i\in\{1,2\}} \alpha_t^i \nabla \log q_t^i(\mathbf{x}_t)\right\|^2 - \sum_{i\in\{1,2\}} \alpha_t^i\|\nabla \log q_t^i(\mathbf{x}_t)\|^2\right] + \sum_{i\in\{1,2\}} \alpha_t^i g_t^i(\mathbf{x}_t).$$

$$\tag{14}$$

Proposition 6.1 and theorem 6.2 are presented in a modular form. This allows us to use arbitrary combinations of operators and propagate the log-weights of components.

## 6.1 INFERENCE PROCEDURE

Together, Definition 4.1, proposition 6.1, and theorem 6.2 define our theoretical basis for arbitrarily nested model composition. During the sampling process, we keep track of the evolution of

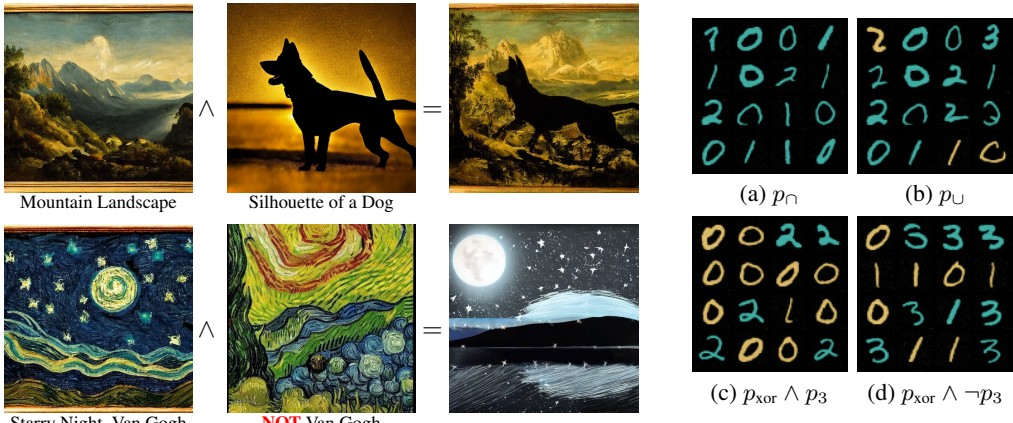

Figure 4: Generated Image Compositions with MNIST ($\lambda \in \{5 \cdot 10^{-3}, 5 \cdot 10^{-2}\}$) and Stable Diffusion ($\lambda = 10$).

loglikelihoods with the Itô density estimator from Equation (3). This efficient density estimation method enables us to perform complex model compositions with minimal overhead. During composition, we can then compose our scores, log-likelihoods, and FKC terms with the procedure described in Algorithm 1. To improve sampling, we can use SMC techniques during the simulation trajectories (Naesseth et al., 2019). In our experiments, we use systematic sampling proportional to the exponentially weighted momentary weight-field $\exp\{g_t(\mathbf{x})dt\}$ (Douc & Cappé, 2005).

---

**Algorithm 1:** DOMBICOMPOSITION over arbitrary formulas

**Input** : scores $\{s^i\}_{i=1}^k$, log-likelihoods $\{\log q^i\}_{i=1}^k$, weights $\{g^i\}_{i=1}^k$, formula $F ::= i|\neg_j i|F_1 \circ F_2$

**Output:** Composite score $s$, Composite log-likelihood $\log q$, Composite weight $g$

1 **if** $F = i$ **then return** $s^i, \log q^i, g^i$

2 **else if** $F = \neg_j i$ **then return** $2s^j - s^i$, $2\log q^j - \log q^i$, $\sigma_t^2 \|s^j - s^i\|^2 + 2g^j - g^i$   // Prop. 6.1

3 **else if** $F = F_1 \wedge_\lambda F_2$ **then** $\lambda \leftarrow -\lambda$       // Conjunction is a *negative* power norm

/* Case $F = F_1 \wedge_\lambda F_2 \mid F_1 \vee_\lambda F_2$:  evaluate subformulas first       */

4 $\bar{s}^1, \overline{\log q}^1, \bar{g}^1 \leftarrow$ DOMBICOMPOSITION($\{s^i\}_{i=1}^k, \{\log q^i\}_{i=1}^k, \{g^i\}_{i=1}^k, F_1$)

5 $\bar{s}^2, \overline{\log q}^2, \bar{g}^2 \leftarrow$ DOMBICOMPOSITION($\{s^i\}_{i=1}^k, \{\log q^i\}_{i=1}^k, \{g^i\}_{i=1}^k, F_2$)

6 $\alpha^1 \leftarrow \text{softmax}_1(\lambda \overline{\log q}^1, \lambda \overline{\log q}^2)$; $\alpha^2 \leftarrow 1 - \alpha^1$

7 $\bar{g} \leftarrow (1-\lambda)\frac{\sigma^2}{2}\left[\|\alpha^1 \bar{s}^1 + \alpha^2 \bar{s}^2\|^2 - (\alpha^1 \|\bar{s}^1\|^2 + \alpha^2 \|\bar{s}^2\|^2)\right]$ // Theorem 6.2

8 **return** $\alpha^1 \bar{s}^1 + \alpha^2 \bar{s}^2$, $\frac{1}{\lambda} LogSumExp(\lambda \overline{\log q}^1, \lambda \overline{\log q}^2)$, $\bar{g} + \alpha^1 \bar{g}^1 + \alpha^2 \bar{g}^2$

---

## 7 EXPERIMENTS

### 7.1 COMBINATORIAL BIAS IN COMPOSITION SAMPLES

We first test the ability of our method to sample from complex compositions of diffusion models *qualitatively* and *quantitatively*. We compose three pretrained models that generate colored MNIST digits (LeCun, 1998). Our three models are defined as follows: Model $p_1$ generates the digits $\{0, 1, 2, 3\}$ in cyan, $p_2$ generates digits smaller 2: $\{0, 1, 0, 1\}$ in cyan or beige and $p_3$ generates the even digits $\{0, 2, 0, 2\}$ in cyan or beige. We would now like to perform set operations on these 7 unique digits-color pairs, similar to Garipov et al. (2023), but with general operations. Figure 4 shows a set of chosen set operations on our models. Beyond the intersection $p_\cap = p_1 \wedge p_2 \wedge p_3$ and the

| Method | Maj$_2$ | | XOR$_2$ | | OneHot$_2$ | | Maj$_{10}$ | | XOR$_{10}$ | | OneHot$_{10}$ | |
|---|---|---|---|---|---|---|---|---|---|---|---|---|
| | Sat↑ | Unif↑ | Sat↑ | Unif↑ | Sat↑ | Unif↑ | Sat↑ | Unif↑ | Sat↑ | Unif↑ | Sat↑ | Unif↑ |
| Dombi | **1.00** | **1.00** | **0.97** | **1.00** | **0.97** | **1.00** | **1.00** | **0.98** | **0.89** | **0.98** | **0.07** | **1.00** |
| PoE/MoE | **1.00** | 0.80 | 0.00 | 0.00 | 0.00 | 0.00 | 0.00 | 0.00 | 0.00 | 0.00 | 0.00 | 0.00 |

Table 1: Quantitative SAT-Experiment, with $2^{14}$ samples from a composition of $k$ models. We report Sat (fraction of particles around satisfying modes), and Unif, the normalized perplexity over satisfying modes (1 = uniform; 0 = collapsed). Higher is better for both metrics.

union $p_{\cup} = p_1 \vee p_2 \vee p_3$ we show results for the exclusive-or operation $p_{\text{xor}} = (p_1 \vee p_2) \wedge (\neg p_1 \vee \neg p_2)$, that samples digits from *either* $p_1$ or $p_2$ but not from their intersection. We then show $p_{\text{xor}} \wedge p_3 = \{2, 0\}$ as well as $p_{\text{xor}} \wedge \neg p_3 = \{3, 1\}$. In all experiments, the reference model for negation is chosen to be the composite score $p_{\cup}$. With few exceptions, we can see that our approach allows us to sample from complex compositions such as $p_{\text{xor}}$ solely via score composition of pretrained diffusion models.

**Quantitative Results**   from a SAT-like toy experiment are shown in Table 1 explained in detail in Appendix E. We model three different composition formulas over $k = \{2, \ldots, 10\}$ models: Maj$_k$ as a *majority*, XOR$_k$ as a formula with an exponential number of operators, One-Hot$_k$ as formula with many negated terms. Compared to PoE/MoE composition, our Dombi composition exhibits less mode-bias (higher Unif.), and is stable (higher SAT.) even in regimes with many negations. We emphasize that Dombi maintains Sat$> 0.9$ even for XOR$_{10}$, which consists of over 1000 score terms.

## 7.2   Multi-Prompt Image Generation and Avoidance

To show the performance of Dombi composition in production-scale diffusion models, we compare its ability to generate images that interpolate between or avoid concepts using Stable Diffusion (SD) v1-4. For all our compositions, we choose two prompts $c_1, c_2$, e.g., `"a mountain landscape"` and `"a silhouette of a dog"`. We then evaluate twenty pairs of images composed conjunctively, as $p(\mathbf{x}|c_1) \wedge p(\mathbf{x}|c_2)$, and compare against and by Skreta et al. (2025b) and scaled PoE, i.e., unweighted averaging of scores (Liu et al., 2022). We further investigate $p(\mathbf{x}|c_1) \wedge \neg_{p(\mathbf{x})} p(\mathbf{x}|c_2)^{\gamma}$ on ten pairs of prompts to illustrate the ability of our model to avoid concepts. As baselines for contrastive prompting, we use ICN (Ho & Salimans, 2021) and the conjunction of (Skreta et al., 2025b), combined with our referenced negation. We use the composed scores in the usual CFG pipeline of SD and measure for all prompts the min. CLIP score (Radford et al., 2021), which measures cosine similarity between image embedding and prompt embedding, and the minimum ImageReward value (Xu et al., 2023), which estimates how closely generated images align with human preferences. For contrastive prompts, we report the difference of each metric between $c1$ and $c2$.

**Dombi Composition**   shows improvement beyond state-of-the-art methods in both CLIP and ImageReward scores, as shown in Tables 5a and 5b, with an example of generated images in Figure 4. For the full list of used prompts, see Appendix D.2. A stark contrast between our method and SuperDiff is evident in Figure 5c, which shows mixture stability over the first 100 iterations of the generation process. The batch variances of the mixture coefficient $\alpha$ are shown to correlate strongly with $\lambda$, with an increase over time caused by different equilibrium points per batch. and (Superdiff) shows strong fluctuations in mixing coefficients, especially during the initial iterations. This effect is more pronounced when we retrofit and to contrastive settings with our negation definition.

## 7.3   Multi-Target Protein Synthesis with FKC Correction

As a final experiment, we test Dombi composition combined with FKC in the setting of structure-based drug design (SBDD). The goal here is to generate molecules (ligands) using the structure of a protein as a guide and evaluate their binding energy (Anderson, 2003). In our experiments, we investigate the impact of FKC from Theorem 6.2 on the quality of Dombi composed results. We generated 32 ligands of sizes $\{15, 19, 23, 27, 35\}$ each, for 14 protein pairs, and evaluated their docking scores using Autodock Vina (Eberhardt et al., 2021) and reproduced the experimental setup of (Skreta et al., 2025a). In this experiment, we use annealing on the base distributions: We evaluate

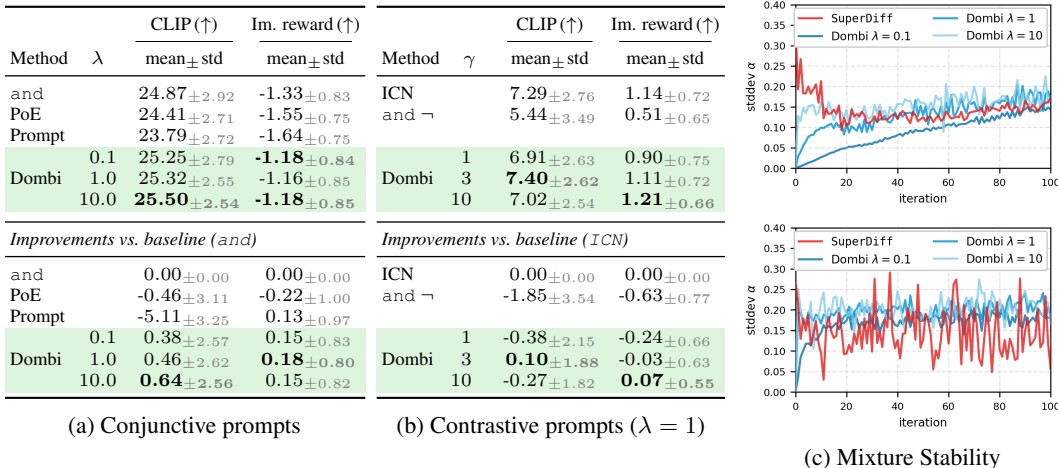

|  |  | CLIP ($\uparrow$) | Im. reward ($\uparrow$) |  |  | CLIP ($\uparrow$) | Im. reward ($\uparrow$) |
|---|---|---|---|---|---|---|---|
| Method | $\lambda$ | mean$_{\pm}$std | mean$_{\pm}$std | Method | $\gamma$ | mean$_{\pm}$std | mean$_{\pm}$std |
| and |  | $24.87_{\pm 2.92}$ | $-1.33_{\pm 0.83}$ | ICN |  | $7.29_{\pm 2.76}$ | $1.14_{\pm 0.72}$ |
| PoE |  | $24.41_{\pm 2.71}$ | $-1.55_{\pm 0.75}$ | and $\neg$ |  | $5.44_{\pm 3.49}$ | $0.51_{\pm 0.65}$ |
| Prompt |  | $23.79_{\pm 2.72}$ | $-1.64_{\pm 0.75}$ |  |  |  |  |
|  | 0.1 | $25.25_{\pm 2.79}$ | $\mathbf{-1.18}_{\pm \mathbf{0.84}}$ |  | 1 | $6.91_{\pm 2.63}$ | $0.90_{\pm 0.75}$ |
| Dombi | 1.0 | $25.32_{\pm 2.55}$ | $-1.16_{\pm 0.85}$ | Dombi | 3 | $\mathbf{7.40}_{\pm \mathbf{2.62}}$ | $1.11_{\pm 0.72}$ |
|  | 10.0 | $\mathbf{25.50}_{\pm \mathbf{2.54}}$ | $-1.18_{\pm 0.85}$ |  | 10 | $7.02_{\pm 2.54}$ | $\mathbf{1.21}_{\pm \mathbf{0.66}}$ |
| *Improvements vs. baseline (and)* |  |  |  | *Improvements vs. baseline (ICN)* |  |  |  |
| and |  | $0.00_{\pm 0.00}$ | $0.00_{\pm 0.00}$ | ICN |  | $0.00_{\pm 0.00}$ | $0.00_{\pm 0.00}$ |
| PoE |  | $-0.46_{\pm 3.11}$ | $-0.22_{\pm 1.00}$ | and $\neg$ |  | $-1.85_{\pm 3.54}$ | $-0.63_{\pm 0.77}$ |
| Prompt |  | $-5.11_{\pm 3.25}$ | $0.13_{\pm 0.97}$ |  |  |  |  |
|  | 0.1 | $0.38_{\pm 2.57}$ | $0.15_{\pm 0.83}$ |  | 1 | $-0.38_{\pm 2.15}$ | $-0.24_{\pm 0.66}$ |
| Dombi | 1.0 | $0.46_{\pm 2.62}$ | $\mathbf{0.18}_{\pm \mathbf{0.80}}$ | Dombi | 3 | $\mathbf{0.10}_{\pm \mathbf{1.88}}$ | $-0.03_{\pm 0.63}$ |
|  | 10.0 | $\mathbf{0.64}_{\pm \mathbf{2.56}}$ | $0.15_{\pm 0.82}$ |  | 10 | $-0.27_{\pm 1.82}$ | $\mathbf{0.07}_{\pm \mathbf{0.55}}$ |

(a) Conjunctive prompts      (b) Contrastive prompts ($\lambda = 1$)      (c) Mixture Stability

Figure 5: Joint generation performance with Stable Diffusion, and paired improvement over baselines with 20 seeds. a shows results for 20 joint prompts $p(\mathbf{x}|c_1) \wedge p(\mathbf{x}|c_2)$. b shows results for 10 contrastive prompts $p(\mathbf{x}|c_1) \wedge \neg_{p(\mathbf{x})} p(\mathbf{x}|c_2)^{\gamma}$. c shows the variance of $\alpha$ during conjunctive (top) and contrastive (bottom) composition. and is from SuperDiff (Skreta et al., 2025b).

| Method | Temp. $\gamma$ | FKC? | ($P_1 * P_2$) ($\uparrow$) | max($P_1, P_2$) ($\downarrow$) | Better than ref. ($\uparrow$) | Div. ($\uparrow$) | Val. & Uniq. ($\uparrow$) | QED ($\uparrow$) | SA ($\downarrow$) |
|---|---|---|---|---|---|---|---|---|---|
| TargetDiff | – | – | $62.19_{\pm 27.08}$ | $-7.24_{\pm 2.35}$ | $0.32_{\pm 0.37}$ | $\mathbf{0.89}_{\pm 0.01}$ | $0.95_{\pm 0.07}$ | $0.57_{\pm 0.14}$ | $\mathbf{0.59}_{\pm 0.09}$ |
| Dombi | 1 | ✗ | $68.60_{\pm 28.09}$ | $-7.42_{\pm 2.57}$ | $0.28_{\pm 0.34}$ | $0.88_{\pm 0.02}$ | $\mathbf{0.96}_{\pm 0.09}$ | $0.58_{\pm 0.13}$ | $0.59_{\pm 0.10}$ |
| Dombi | 1 | ✓ | $72.83_{\pm 22.42}$ | $-7.71_{\pm 1.65}$ | $0.27_{\pm 0.35}$ | $0.86_{\pm 0.03}$ | $0.95_{\pm 0.08}$ | $0.57_{\pm 0.13}$ | $0.59_{\pm 0.11}$ |
| Dombi | 2 | ✗ | $71.36_{\pm 29.44}$ | $-7.59_{\pm 2.48}$ | $0.30_{\pm 0.34}$ | $0.88_{\pm 0.01}$ | $0.93_{\pm 0.16}$ | $\mathbf{0.59}_{\pm 0.12}$ | $0.62_{\pm 0.09}$ |
| Dombi | 2 | ✓ | $\mathbf{81.63}_{\pm 25.91}$ | $\mathbf{-8.25}_{\pm 1.56}$ | $\mathbf{0.38}_{\pm 0.40}$ | $0.85_{\pm 0.11}$ | $0.93_{\pm 0.17}$ | $\mathbf{0.59}_{\pm 0.12}$ | $0.62_{\pm 0.10}$ |

Table 2: Docking Scores of generated ligands for 14 protein target pairs ($P_1$, $P_2$), in batches of 32 ligands for 5 molecule lengths each. We compare conjunction with Dombi ($\lambda = 1$) with and without FKC with annealed base distribution and also report TargetDiff from (Guan et al., 2023) as baseline.

$p(\mathbf{x}|P_1)^{\gamma} \wedge p(\mathbf{x}|P_2)^{\gamma}$, and propagate the FKC term of the annealed base distributions to our dombi operator as in Algorithm 1. Per batch, we report the average joint docking performance to each target protein as their product ($P_1 * P_2$), the objective of PoE, as well as max($P_1, P_2$), which is closer to the objective of the Dombi composition. Further, we measure the fraction of molecules that have a higher docking score than the known reference molecules, the diversity of molecules, as well as the fraction of valid *and* unique molecules, and their drug-likeness (QED) (Bickerton et al., 2012) and how easy they are to synthesize (SA) (Ertl & Schuffenhauer, 2009).

**FKC Correction** improves the docking performance in annealed and unannealed settings, as shown in Table 2. The difference is more pronounced for $\gamma = 2$, where we also collect FKC terms for the annealed base distributions. In Appendix D.3 we show results with an additional small sweep over $\lambda$ values, where the performance for $\lambda = 0.3$ and $\lambda = 3$ shows to be similar.

## 8 CONCLUSION AND FUTURE WORK

In this work, we introduced Dombi composition operators as a purely online, well-defined, general class of score-composition operators. Based on power norms, our method recovers and unifies prior work, like MoE, the harmonic mean, or contrast operators (Garipov et al., 2023), yet offers theoretical benefits that are crucial to ensure stability when score compositions become more complex. An important future direction is dynamic schedules for $\lambda$, as Proposition 5.2 suggests that adaptive choices depending on the scores might be better suited to ensure stability. This work opens up some exciting possibilities, e.g., potential applications in neurosymbolic methods, where modular diffusion models could be coupled to solve combinatorial tasks. Furthermore, the option to rewrite formulas might be utilized to switch to different sampling techniques for, e.g., factoring out subformulas.

## REPRODUCIBILITY STATEMENT

Detailed proofs are provided in the Appendix for all our theoretical results. We also provide a link to a GitHub repository containing all the code used to reproduce the results in this manuscript[1]. The repository contains the details required to reproduce the empirical results, including our hyperparameter settings. We make our code public under the MIT License.

## ETHICS STATEMENT

We provide here a principled framework for composing pretrained diffusion models. Diffusion models can be composed to enable generative capabilities beyond what the individual models can provide. While the harmful effects of the misuse of composition cannot be ruled out, our work does not raise any further specific ethical concerns that need to be highlighted here.

## ACKNOWLEDGMENTS

The authors thank the anonymous reviewer 9bJN for helpful suggestions and discussion. VG acknowledges the support from the Research Council of Finland (grant 342077), Saab-WASP (grant 411025), and the Jane and Aatos Erkko Foundation (grant 7001703).

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

Figure 6: Overview of the main contributions in this work.

## A  OVERVIEW OF THE PAPER

We provide a graphic to summarize the structure of this paper and our main contributions in Figure 1, and restate this in a more structured way in Figure 6.

## B  DE MORGAN DUAL DENSITY OPERATORS

In this section, we provide a general method for deriving a score calculus with the desired **pointwise** properties over the underlying (possibly unnormalized) probability densities. As described in Section 2.3, the semantics we seek align with "fuzzy intersection/conjunction" operators $\wedge$, more formally known as t-norms (Klement et al., 2013). To obtain the "fuzzy union/disjunction" operator $\vee$ corresponding to $\wedge$, we choose to use De Morgan's law: $x \vee y := \neg(\neg x \wedge \neg y)$. Here, $\neg$ is a negation operator. Choosing a De Morgan dual pair of operators allows for semantics-preserving rewrites (e.g., into negation normal form), which is convenient for implementing large compositions.

To connect theory in fuzzy logic to our setting of density operators, we require a lifting function to map the target domain of densities, $\mathbb{R}_{\geq 0} \cup \{\infty\}$, which we will denote with $[0, \infty]$, to the domain of memberships, $[0, 1]$. We use a lifting function $\phi : [0, \infty] \to [0, 1]$ to accomplish this. As $\phi$ should maintain the semantics of high densities corresponding to high densities, we define it to be an order-isomorphism. Finally, we can use a result of (Dombi, 1982) to define a class of De Morgan Dual operators on (unnormalized) probability densities.

Later in this section and in the main text, we implement these density operators through their induced score calculus, since diffusion samplers consume score fields rather than normalized densities.

**Definition B.1** (De Morgan Dual Density Operators). *Let $\phi : [0, \infty] \to [0, 1]$ be an order-isomorphism and $f : [0, 1] \to [0, \infty]$ be a continuous, strictly decreasing function with $f(0) = \infty$. For $g = f \circ \phi$, we define for density function $p$*

$$\neg p(\mathbf{x}) := \phi^{-1}(1 - \phi(p(\mathbf{x}))) \tag{15}$$

$$p_1(\mathbf{x}) \wedge p_2(\mathbf{x}) := g^{-1}(g(p_1(\mathbf{x})) + g(p_2(\mathbf{x}))) \tag{16}$$

$$p_1(\mathbf{x}) \vee p_2(\mathbf{x}) := \neg(\neg p_1(\mathbf{x}) \wedge \neg p_2(\mathbf{x})) \tag{17}$$

By construction, we have the following properties.

**Proposition B.2** (Algebraic properties (pointwise)). *Assume $\phi : [0, \infty] \to [0, 1]$ is a continuous order-isomorphism, and assume $g = f \circ \phi : [0, \infty] \to [0, \infty]$ is a bijection with $g(\infty) = 0$. Define $\neg, \wedge, \vee$ as Definition B.1. Then for all $u, v, w \in [0, \infty]$:*

- *$([0, \infty], \wedge, \infty)$ is a commutative monoid.*

- *$([0, \infty], \vee, 0)$ is a commutative monoid.*

- $\neg$ *is involutive:* $\neg(\neg u) = u$.

- *(De Morgan)* $\neg(u \wedge v) = \neg u \vee \neg v$ *and* $\neg(u \vee v) = \neg u \wedge \neg v$.

- *(Neutral Element) There exists a unique* $u^* \in [0, \infty]$ *with* $\neg u^* = u^*$, *given by* $u^* = \phi^{-1}(1/2)$.

## B.1 DE MORGAN DUAL SCORE OPERATORS

As Diffusion models operate on scores, i.e., gradients of log-likelihoods, we need to transfer our operators to the score domain. We use the composite generator function $g : f \circ \phi$ from Definition B.1, and sketch that De Morgan Dual Density Connectives $\wedge, \vee$ translate to weighted score combinations.

**Lemma B.3** (De Morgan Dual Score Operators are linear combinations). *Let* $\phi$ *and* $f$ *be fully differentiable and defined as in Definition B.1. We again let* $g = f \circ \phi$. *For any two strictly positive density functions* $p_1, p_2$ *with* $q = p_1 \wedge p_2 = g^{-1}(g(p_1) + g(p_2))$, *it holds pointwise that*

$$\nabla \log q = \alpha_1 \nabla \log p_1 + \alpha_2 \nabla \log p_2, \tag{18}$$

*where the scalar weights* $\alpha_i := w(p_i)/w(p_1 \wedge p_2)$, *with the weighting function* $w(u) := ug'(u)$.

*Importantly,* $\alpha_i$ *are dynamic "responsibility weights".*

*Proof.* The statement follows from standard calculation

$$g(q) = g(p_1) + g(p_2) \tag{19}$$

$$g'(q)\nabla q = g'(p_1)\nabla p_1 + g'(p_2)\nabla p_2 \tag{20}$$

$$w(q)\frac{\nabla q}{q} = w(p_1)\frac{\nabla p_1}{p_1} + w(p_2)\frac{\nabla p_2}{p_2} \tag{21}$$

$$w(q)\nabla \log q = w(p_1)\nabla \log p_1 + w(p_2)\nabla \log p_2 \tag{22}$$

$$\nabla \log q = \alpha_1 \nabla \log p_1 + \alpha_2 \nabla \log p_2 \tag{23}$$

$\square$

The statement for $\vee$ follows analogously, and for negations, we can always rewrite for some density-dependent factor $\kappa$ $\nabla \log(\neg p) = \kappa(p)\nabla \log p$. This, together with De Morgan's law, allows us to represent *any* nested formula over scores as a linear combination over (negated) base scores.

Lemma B.3 allows us to extend the properties in Proposition B.2 to score models.

**Proposition B.4** (De Morgan Dual Score operators are linear combinations). *Let* $F$ *be any composition function obtained by nested application of* $\wedge, \vee, \neg$ *over density functions* $p_1, \ldots, p_k$. *Then there exist scalar weights* $\alpha_i$ *such that*

$$\nabla \log F(p_1, \ldots, p_k) = \sum_{i=1}^{k} \alpha_i \nabla \log p_i \tag{24}$$

*Proof.* The result follows from the application of De Morgan's law to rewrite $F$ into negated normal form, i.e., as an otherwise negation-free formula over (negated) literals. Then, $\alpha_i$ can be computed by repeated application of Lemma B.3, and positive and negative occurrences are collected together with $\kappa(p_i) := \frac{p_i h'(p_i)}{h(p_i)}$ and $h(p_i) := \phi^{-1}(1 - \phi(p_i))$ as $\nabla \log \neg p_i = \kappa(p_i)\nabla \log p_i$. $\square$

## B.2 REFERENCED NEGATION AND CONDITIONAL GENERATION

As we care specifically about the pointwise properties of our operators, we can let the generators $f$ and $\phi$ themselves vary. In particular, we may use $\phi(p; \mathbf{x})$ as a location-dependent lifting function. This is particularly useful to draw parallels to related literature in guided generation/avoidance. The choice of $\phi$ defines our fixed point with respect to negation, and a semantically meaningful fixed point can be modelled with a *unconditional base model*.

In conditional generation, we are interested in operations on the conditional distributions (Du et al., 2020; Liu et al., 2022). More formally, we are given conditional models $p(\mathbf{x} \mid y_i)$ for given *concepts* $y_i$, as well as the unconditional model $p(\mathbf{x})$ and are interested in sampling with a formula $F$ from $p(\mathbf{x} \mid F(y_1, \ldots, y_k))$. In the typical probabilistic interpretations, (Du & Kaelbling, 2024) implement these operations on *density ratios*. This, in our setting, would translate to

$$p(\mathbf{x} \mid F(y_1, \ldots, y_k)) \propto p(\mathbf{x}, F(y_1, \ldots, y_k)) = p(\mathbf{x}) F\left( \frac{p(\mathbf{x} \mid y_1)}{p(\mathbf{x})}, \ldots, \frac{p(\mathbf{x} \mid y_k)}{p(\mathbf{x})} \right). \tag{25}$$

As the denominator $p(\mathbf{x})$ is constant across the formula, we may absorb it into $\phi$ as *constant negation fixed-point*. We then denote in the notation of the main paper with $\phi_c(p; \mathbf{x}) := \phi\left( \frac{p(\mathbf{x})}{c(\mathbf{x})} \right)$, where now $p(\mathbf{x})$ takes the role of the conditional distribution, and $c(\mathbf{x})$ is the constant negation fixed-point.

### B.3 DERIVATION AND PROPERTIES OF DOMBI CALCULUS

In the main paper, we focus on a particular class of De Morgan dual density operators: the Dombi operators. We follow the definition of Dombi (1982), lifted to our setting, to define the Dombi operators with $\phi_c(x) = \frac{x}{x+c} = \frac{1}{\frac{c}{x}+1}$ and $f_\lambda(x) = \left( \frac{1}{x} - 1 \right)^\lambda$, and derive their corresponding score calculus here. First, we can see here that $\phi_c^{-1}(x) = \frac{cx}{1-x} = \frac{c}{\frac{1}{x}-1}$, $g(x) = f_\lambda(\phi_c(x)) = \left( \frac{c}{x} \right)^\lambda$, $h(x) = f_\lambda(1 - \phi_c(x)) = f_\lambda\left( \frac{c}{x+c} \right) = f_\lambda\left( \frac{1}{\frac{x}{c}+1} \right) = \left( \frac{x}{c} \right)^\lambda$. Further $g^{-1}(x) = cx^{-1/\lambda}$. With this we can derive Definition 4.1 as:

$$\neg p(\mathbf{x}) = \phi_c^{-1}(1 - \phi_c(p(\mathbf{x}))) = \phi_c^{-1}\left( \frac{c(\mathbf{x})}{c(\mathbf{x}) + p(\mathbf{x})} \right) = \frac{c(\mathbf{x})^2}{p(\mathbf{x})} \tag{26}$$

$$p_1(\mathbf{x}) \wedge_\lambda p_2(\mathbf{x}) := g^{-1}(g(p_1(\mathbf{x})) + g(p_2(\mathbf{x}))) \tag{27}$$

$$= c(\mathbf{x}) \left( \left( \frac{c(\mathbf{x})}{p_1(\mathbf{x})} \right)^\lambda + \left( \frac{c(\mathbf{x})}{p_2(\mathbf{x})} \right)^\lambda \right)^{-1/\lambda} \tag{28}$$

$$= \left( \left( \frac{1}{p_1(\mathbf{x})} \right)^\lambda + \left( \frac{1}{p_2(\mathbf{x})} \right)^\lambda \right)^{-1/\lambda} \tag{29}$$

$$= \left( p_1(\mathbf{x})^{-\lambda} + p_2(\mathbf{x})^{-\lambda} \right)^{-1/\lambda} \tag{30}$$

$$p_1(\mathbf{x}) \vee_\lambda p_2(\mathbf{x}) := \neg_c(\neg_c p_1(\mathbf{x}) \wedge_\lambda \neg_c p_2(\mathbf{x})) \tag{31}$$

$$= \frac{c(\mathbf{x})^2}{\frac{c(\mathbf{x})^2}{p_1(\mathbf{x})} \wedge_\lambda \frac{c(\mathbf{x})^2}{p_2(\mathbf{x})}} \tag{32}$$

$$= \frac{1}{\frac{1}{p_1(\mathbf{x})} \wedge_\lambda \frac{1}{p_2(\mathbf{x})}} \tag{33}$$

$$= \frac{1}{\left( p_1(\mathbf{x})^\lambda + p_2(\mathbf{x})^\lambda \right)^{-1/\lambda}} \tag{34}$$

$$= \left( p_1(\mathbf{x})^\lambda + p_2(\mathbf{x})^\lambda \right)^{1/\lambda} \tag{35}$$

In log-likelihoods and scores, the negation is straightforward. For a power-mixture $\left( p_1(\mathbf{x})^\lambda + p_2(\mathbf{x})^\lambda \right)^{1/\lambda}$, the log-likelihood and score operations are familiar. We investigate disjunction and conjunction at the same time and state for all $\lambda \neq 0$:

$$q(\mathbf{x}) = \left(p_1(\mathbf{x})^\lambda + p_2(\mathbf{x})^\lambda\right)^{1/\lambda} \qquad\qquad \Longrightarrow \qquad (36)$$

$$\log q(\mathbf{x}) = \frac{1}{\lambda}\log\left(p_1(\mathbf{x})^\lambda + p_2(\mathbf{x})^\lambda\right) \qquad\qquad (37)$$

$$\frac{1}{\lambda}\log\left(\exp(\lambda\log p_1(\mathbf{x})) + \exp(\lambda\log p_2(\mathbf{x}))\right) \qquad\qquad (38)$$

$$\frac{1}{\lambda}\mathrm{LogSumExp}(\lambda\log p_1(\mathbf{x}), \lambda\log p_2(\mathbf{x})) \qquad\qquad \Longrightarrow \qquad (39)$$

$$\nabla_{\mathbf{x}}\log q(\mathbf{x}) = \sum_{i\in\{1,2\}}\left(\mathrm{softmax}_i(\lambda\log p_1(\mathbf{x}), \lambda\log p_2(\mathbf{x})\nabla_{\mathbf{x}}\log p_i(\mathbf{x})\right) \qquad (40)$$

$$= \sum_{i\in\{1,2\}}\left(\frac{p_i(\mathbf{x})^\lambda}{p_1(\mathbf{x})^\lambda + p_2(\mathbf{x})^\lambda}\nabla_{\mathbf{x}}\log p_i(\mathbf{x})\right) \qquad (41)$$

In terms of score calculus, the Dombi Operators end up being a softmax-weighted, convex combination of the component scores, in accordance with Lemma B.3. Further, they inherit the favorable properties in Proposition B.2.

### B.4 DOMBI ERROR BOUNDS

For a given value of $\lambda$, the maximal difference between the Dombi operators and the $\min/\max$ functions can be easily bounded as an additive term in log-likelihood:

**Proposition B.5.** *Let $\wedge_\lambda, \vee_\lambda$ be the Dombi density operators. Then it holds that*

$$\forall x, y \in \mathbb{R}_{\geq 0}: \qquad \min\{x,y\}2^{-1/\lambda} \leq x \wedge_\lambda y \leq \min\{x,y\} \qquad (42)$$

$$\forall x, y \in \mathbb{R}_{\geq 0}: \qquad \max\{x,y\} \leq x \vee_\lambda y \leq \max\{x,y\}2^{1/\lambda} \qquad (43)$$

*Proof.* See Appendix C  $\qquad\qquad\square$

## C PROOFS

**Proposition B.5.** *Let $\wedge_\lambda, \vee_\lambda$ be the Dombi density operators. Then it holds that*

$$\forall x, y \in \mathbb{R}_{\geq 0}: \qquad \min\{x,y\}2^{-1/\lambda} \leq x \wedge_\lambda y \leq \min\{x,y\} \qquad (42)$$

$$\forall x, y \in \mathbb{R}_{\geq 0}: \qquad \max\{x,y\} \leq x \vee_\lambda y \leq \max\{x,y\}2^{1/\lambda} \qquad (43)$$

*Proof.* We show the case for $p \vee_\lambda q = \left(p^\lambda + q^\lambda\right)^{1/\lambda}$ first. The definition of $\vee_\lambda$ is equivalent to that of a P-norm over two components. We have the standard inequality (w.l.o.g. for $p \geq q$)

$$p \vee_\lambda q = \left(p^\lambda + q^\lambda\right)^{1/\lambda} \leq \left(2p^\lambda\right)^{1/\lambda} = 2^{1/\lambda}\max\{p,q\} \qquad (44)$$

The lower bound similarly follows from

$$p \vee_\lambda q = \left(p^\lambda + q^\lambda\right)^{1/\lambda} \geq \left(p^\lambda\right)^{1/\lambda} = \max\{p,q\} \qquad (45)$$

We can note that the upper bound is tight for $p = q$ and the lower bound is tight for $q = 0$. For $\wedge_\lambda$, we can use De Morgan to obtain the symmetric bounds.

$\qquad\qquad\square$

**Proposition 5.2** (Mixture Stability)**.** *Let $\alpha_t = \mathrm{softmax}_1(\lambda\log p_t^1, \lambda\log p_t^2)$, for a Dombi composition $p^1 \circ_\lambda p^2$. Then it holds for the score difference $\Delta s = s_t^1 - s_t^2$*

$$|\mathbb{E}[d\alpha_t \mid \mathbf{x}_t]| = \mathcal{O}(|\lambda^2 - \lambda|\sigma_t^2\|\Delta s\|^2 dt), \qquad\qquad d[\alpha]_t = \mathcal{O}(\lambda^2\sigma_t^2\|\Delta s\|^2 dt)$$

*Proof.* Our objective is to bound the rate of change of the stochastic mixing weight $\alpha$. We use $\ell = \log p_t^1 - \log p_t^2$ and $s = \alpha s^1 + (1-\alpha)s^2, \Delta s = s^1 - s^2$ as shorthands for this proof.

First, we compute $d\ell$ and then use this to bound the rate of change for $\alpha = \text{sigmoid}(\lambda\ell)$. By Equation (3), we have for $u = -f_t + \sigma^2 s$

$$d\ell = \langle dx, \Delta s \rangle + \langle \Delta s, f_t \rangle \, dt - \frac{\sigma_t^2}{2} \left( \|s_t^1\|^2 - \|s_t^2\|^2 \right) dt \tag{46}$$

$$= \langle \Delta s, u \rangle \, dt + \sigma_t \langle \Delta s, d\overline{\mathbf{w}} \rangle + \langle \Delta s, f_t \rangle \, dt - \frac{\sigma_t^2}{2} \left( \|s_t^1\|^2 - \|s_t^2\|^2 \right) dt \tag{47}$$

$$= \sigma_t^2 \langle \Delta s, s \rangle \, dt - \frac{\sigma_t^2}{2} \left( \|s_t^1\|^2 - \|s_t^2\|^2 \right) dt + \sigma_t \langle \Delta s, d\overline{\mathbf{w}} \rangle \tag{48}$$

$$= \frac{\sigma_t^2}{2} \left\langle \Delta s, \ 2s - (s_t^1 + s_t^2) \right\rangle dt + \sigma_t \langle \Delta s, d\overline{\mathbf{w}} \rangle. \tag{49}$$

And, for later convenience, we explicitly state that $d[\ell]_t = \sigma_t^2 \|\Delta s\|^2 dt$.

We now apply Ito's Lemma $d\,\phi(\ell_t) = \phi'(\ell_t)\,d\ell_t + \frac{1}{2}\phi''(\ell_t)\,d[\ell]_t$ to $\phi(\ell) := \text{sigmoid}(\lambda\ell)$ and get

$$d\alpha = \lambda\alpha(1-\alpha)\,d\ell + \frac{1}{2}\,\lambda^2\alpha(1-\alpha)(1-2\alpha)\,d[\ell]. \tag{50}$$

Finally, we combine terms to

$$d\alpha = \lambda\alpha(1-\alpha)\left( \frac{\sigma_t^2}{2} \left\langle \Delta s, \ 2s - (s_t^1 + s_t^2) \right\rangle dt + \sigma_t \langle \Delta s, d\overline{\mathbf{w}} \rangle \right) + \frac{1}{2}\,\lambda^2\alpha(1-\alpha)(1-2\alpha)\,\sigma_t^2 \|\Delta s\|^2\,dt \tag{51}$$

$$= \lambda\alpha(1-\alpha)\,\sigma_t \langle \Delta s, d\overline{\mathbf{w}} \rangle + \frac{\sigma_t^2}{2}\alpha(1-\alpha)\left( \lambda\langle \Delta s, \ 2s - (s_t^1 + s_t^2) \rangle + \lambda^2(1-2\alpha)\|\Delta s\|^2 \right) dt \tag{52}$$

$$= \lambda\alpha(1-\alpha)\,\sigma_t \langle \Delta s, d\overline{\mathbf{w}} \rangle + \frac{\sigma_t^2}{2}\alpha(1-\alpha)(1-2\alpha)(\lambda^2 - \lambda)\,\|\Delta s\|^2\,dt \tag{53}$$

where $\max_{\alpha \in [0,1]} |\alpha(1-\alpha)(1-2\alpha)| = \frac{\sqrt{3}}{18}$. Hence, the drift satisfies

$$|\mathbb{E}[d\alpha_t \mid \mathbf{x}_t]| \le \frac{\sigma_t^2 \sqrt{3}}{36}\,|\lambda^2 - \lambda|\,\|\Delta s\|^2 dt,$$

and the quadratic variation satisfies

$$d[\alpha]_t = \lambda^2\alpha^2(1-\alpha)^2\sigma_t^2\|\Delta s\|^2\,dt \le \frac{\lambda^2\sigma_t^2}{16}\|\Delta s\|^2\,dt.$$

$\square$

## C.1 FEYNMAN-KAC CORRECTION

The reweighting equation $dw_t = \overline{g}(\mathbf{x})dt \implies \frac{\partial p_t(\mathbf{x})}{\partial t} = \overline{g}_t(\mathbf{x})p_t(\mathbf{x})$ describes how the log-weight-field influences the marginals of the weighted SDE. The translation of continuity (drift) terms and diffusion terms into log-weights is then given by the following schemes:

$$\frac{\partial p_t(\mathbf{x})}{\partial t} = -\langle \nabla, p_t(\mathbf{x})v_t(\mathbf{x}) \rangle = \left( \frac{-1}{p_t(\mathbf{x})} \langle \nabla, p_t(\mathbf{x})v_t(\mathbf{x}) \rangle \right) p_t(\mathbf{x}) \implies$$
$$dw_t = \left( -\langle \nabla, v_t(\mathbf{x}) \rangle - \langle \nabla \log p_t(\mathbf{x}), v_t(\mathbf{x}) \rangle \right) \tag{54}$$

for drift terms and

$$\frac{\partial p_t(\mathbf{x})}{\partial t} = \frac{\sigma^2}{2}\Delta p_t(\mathbf{x}) = \frac{\sigma^2}{2}p_t(\mathbf{x})\left( \Delta \log p_t(\mathbf{x}) + \|\nabla \log p_t(\mathbf{x})\|^2 \right) \implies$$
$$dw_t = \frac{\sigma^2}{2}\left( \Delta \log p_t(\mathbf{x}) + \|\nabla \log p_t(\mathbf{x})\|^2 \right) \tag{55}$$

for diffusion terms.

Dombi Composition is equivalent to applying a power-norm to probability distributions. We recast this as annealing, a case shown by Skreta et al. (2025a), then taking an (unweighted) mixture and then inverse annealing of the mixture of annealed distributions. We state the following results before proceeding with the main proofs.

**Lemma C.1** (Mixture of SDEs + FKC). *Consider two weighted diffusion models $q_t^1(\mathbf{x}), q_t^2(\mathbf{x})$ defined via the Feynman-Kac equation with corresponding weights $g_t^1(\mathbf{x}), g_t^2(\mathbf{x})$. The weighted SDE corresponding to the sum of the marginals $p_t(\mathbf{x}) \propto q_t^1(\mathbf{x}) + q_t^2(\mathbf{x})$, with $\alpha_t^i = \frac{q_t^i(\mathbf{x})}{q_t^1(\mathbf{x})+q_t^2(\mathbf{x})} \in (0,1)$*

$$
\begin{aligned}
d\mathbf{x}_t &= \left[-f_t(\mathbf{x}_t) + \sigma_t^2(\alpha_t^1 \nabla \log q_t^1(\mathbf{x}_t) + \alpha_t^2 \nabla \log q_t^2(\mathbf{x}_t))\right] dt + \sigma_t d\overline{\mathbf{w}}_t \\
dw_t &= \left[\alpha_t^1 g_t^1(\mathbf{x}) + \alpha_t^2 g_t^2(\mathbf{x})\right] dt
\end{aligned}
\tag{56}
$$

*Proof.* We have, for $\bar{g}_t(\mathbf{x}) = \alpha_t^1 \bar{g}_t^1(\mathbf{x}) + \alpha_t^2 \bar{g}_t^2(\mathbf{x})$

$$
\frac{\partial p_t}{\partial t} = \frac{\partial q_t^1}{\partial t} + \frac{\partial q_t^2}{\partial t} - \int \frac{\partial q_t^1}{\partial t} + \frac{\partial q_t^2}{\partial t} d\mathbf{x}
\tag{57}
$$

$$
\begin{aligned}
= &-\langle \nabla, q_t^1(\mathbf{x})(-f_t + \sigma_t^2 \nabla \log q_t^1(\mathbf{x}))\rangle + \frac{\sigma_t^2}{2}\Delta q_t^1(\mathbf{x}) + q_t^1(\mathbf{x})\left[\bar{g}_t^1(\mathbf{x})\right] + \\
&-\langle \nabla, q_t^2(\mathbf{x})(-f_t + \sigma_t^2 \nabla \log q_t^2(\mathbf{x}))\rangle + \frac{\sigma_t^2}{2}\Delta q_t^2(\mathbf{x}) + q_t^2(\mathbf{x})\left[\bar{g}_t^2(\mathbf{x})\right] - \int \frac{\partial q_t^1}{\partial t} + \frac{\partial q_t^2}{\partial t} d\mathbf{x}
\end{aligned}
\tag{58}
$$

$$
\begin{aligned}
= &-\langle \nabla, q_t^1(\mathbf{x})(-f_t + \sigma_t^2 \frac{1}{q_t^1(\mathbf{x})}\nabla q_t^1(\mathbf{x}))\rangle + \frac{\sigma_t^2}{2}\Delta q_t^1(\mathbf{x}) + q_t^1(\mathbf{x})\left[\bar{g}_t^1(\mathbf{x})\right] + \\
&-\langle \nabla, q_t^2(\mathbf{x})(-f_t + \sigma_t^2 \frac{1}{q_t^2(\mathbf{x})}\nabla q_t^2(\mathbf{x}))\rangle + \frac{\sigma_t^2}{2}\Delta q_t^2(\mathbf{x}) + q_t^2(\mathbf{x})\left[\bar{g}_t^2(\mathbf{x})\right] - \int \frac{\partial q_t^1}{\partial t} + \frac{\partial q_t^2}{\partial t} d\mathbf{x}
\end{aligned}
\tag{59}
$$

$$
\begin{aligned}
= &-\langle \nabla, q_t^1(\mathbf{x})(-f_t + \sigma_t^2 \frac{1}{q_t^1(\mathbf{x})}\nabla q_t^1(\mathbf{x})) + q_t^2(\mathbf{x})(-f_t + \sigma_t^2 \frac{1}{q_t^2(\mathbf{x})}\nabla q_t^2(\mathbf{x}))\rangle + \\
&\frac{\sigma_t^2}{2}\Delta p_t(\mathbf{x}) + p_t(\mathbf{x})\bar{g}_t(\mathbf{x}) - \int \frac{\partial p_t}{\partial t} d\mathbf{x}
\end{aligned}
\tag{60}
$$

$$
\begin{aligned}
= &-\langle \nabla, (q_t^1(\mathbf{x}) + q_t^2(\mathbf{x}))(-f_t) + q_t^1(\mathbf{x})(\sigma_t^2 \frac{1}{q_t^1(\mathbf{x})}\nabla q_t^1(\mathbf{x})) + q_t^2(\mathbf{x})(\sigma_t^2 \frac{1}{q_t^2(\mathbf{x})}\nabla q_t^2(\mathbf{x}))\rangle + \\
&\frac{\sigma_t^2}{2}\Delta p_t(\mathbf{x}) + p_t(\mathbf{x})\bar{g}_t(\mathbf{x}) - \int \frac{\partial p_t}{\partial t} d\mathbf{x}
\end{aligned}
\tag{61}
$$

$$
\begin{aligned}
= &-\langle \nabla, (q_t^1(\mathbf{x}) + q_t^2(\mathbf{x}))(-f_t) + (\sigma_t^2 \nabla(q_t^1(\mathbf{x}) + q_t^2(\mathbf{x})))\rangle + \\
&\frac{\sigma_t^2}{2}\Delta p_t(\mathbf{x}) + p_t(\mathbf{x})\bar{g}_t(\mathbf{x}) - \int \frac{\partial p_t}{\partial t} d\mathbf{x}
\end{aligned}
\tag{62}
$$

$$
\begin{aligned}
= &-\langle \nabla, (q_t^1(\mathbf{x}) + q_t^2(\mathbf{x}))(-f_t) + p_t(\mathbf{x})\left(\sigma_t^2 \left(\frac{\nabla q_t^1(\mathbf{x})}{p_t(\mathbf{x})} + \frac{\nabla q_t^2(\mathbf{x})}{p_t(\mathbf{x})}\right)\right)\rangle + \\
&\frac{\sigma_t^2}{2}\Delta p_t(\mathbf{x}) + p_t(\mathbf{x})\bar{g}_t(\mathbf{x}) - \int \frac{\partial p_t}{\partial t} d\mathbf{x}
\end{aligned}
\tag{63}
$$

$$
\begin{aligned}
= &-\langle \nabla, p_t(\mathbf{x})(-f_t) + p_t(\mathbf{x})\left(\sigma_t^2 \left(\frac{\nabla q_t^1(\mathbf{x})}{p_t(\mathbf{x})} + \frac{\nabla q_t^2(\mathbf{x})}{p_t(\mathbf{x})}\right)\right)\rangle + \\
&\frac{\sigma_t^2}{2}\Delta p_t(\mathbf{x}) + p_t(\mathbf{x})\bar{g}_t(\mathbf{x}) - \int \frac{\partial p_t}{\partial t} d\mathbf{x}
\end{aligned}
\tag{64}
$$

$$
= -\langle \nabla, p_t(\mathbf{x})\left(-f_t + \sigma_t^2 \left(\frac{\nabla q_t^1(\mathbf{x})}{p_t(\mathbf{x})} + \frac{\nabla q_t^2(\mathbf{x})}{p_t(\mathbf{x})}\right)\right)\rangle + \frac{\sigma_t^2}{2}\Delta p_t(\mathbf{x}) + p_t(\mathbf{x})\bar{g}_t(\mathbf{x}) - \int \frac{\partial p_t}{\partial t} d\mathbf{x}
\tag{65}
$$

$$
\begin{aligned}
= &-\langle \nabla, p_t(\mathbf{x})\left(-f_t + \sigma_t^2 \left(\frac{q_t^1(\mathbf{x})}{p_t(\mathbf{x})}\nabla \log q_t^1(\mathbf{x}) + \frac{q_t^2(\mathbf{x})}{p_t(\mathbf{x})}\nabla \log q_t^2(\mathbf{x})\right)\right)\rangle + \\
&\frac{\sigma_t^2}{2}\Delta p_t(\mathbf{x}) + p_t(\mathbf{x})\bar{g}_t(\mathbf{x}) - \int \frac{\partial p_t}{\partial t} d\mathbf{x}
\end{aligned}
\tag{66}
$$

$$
\begin{aligned}
= &-\langle \nabla, p_t(\mathbf{x})\left(-f_t + \sigma_t^2 \left(\alpha_t^1 \nabla \log q_t^1(\mathbf{x}) + \alpha_t^2 \nabla \log q_t^2(\mathbf{x})\right)\right)\rangle + \\
&\frac{\sigma_t^2}{2}\Delta p_t(\mathbf{x}) + p_t(\mathbf{x})\bar{g}_t(\mathbf{x}) - \int \frac{\partial p_t}{\partial t} d\mathbf{x}
\end{aligned}
\tag{67}
$$

$$
= -\langle \nabla, p_t(\mathbf{x})\left(-f_t + \sigma_t^2 \left(\alpha_t^1 \nabla \log q_t^1(\mathbf{x}) + \alpha_t^2 \nabla \log q_t^2(\mathbf{x})\right)\right)\rangle + \frac{\sigma_t^2}{2}\Delta p_t(\mathbf{x}) + p_t(\mathbf{x})\bar{g}_t(\mathbf{x}) - 0
\tag{68}
$$

We can simulate this as

$$
\begin{aligned}
d\mathbf{x}_t &= \left[ -f_t(\mathbf{x}_t) + \sigma_t^2(\alpha_t^1 \nabla \log q_t^1(\mathbf{x}_t) + \alpha_t^2 \nabla \log q_t^2(\mathbf{x}_t)) \right] dt + \sigma_t d\overline{\mathbf{w}}_t \\
dw_t &= \left[ \alpha_t^1 g_t^1(\mathbf{x}) + \alpha_t^2 g_t^2(\mathbf{x}) \right] dt
\end{aligned}
\tag{69}
$$

$\square$

**Lemma C.2** (Target Score Annealed SDE + FKC, Skreta et al., 2025a). *Consider a diffusion model $q_t(\mathbf{x})$ defined via the Feynman-Kac equation with the weight-field $g_t(\mathbf{x})$ and some parameter $\lambda \in \mathbb{R} \setminus \{0\}$. The weighted SDE corresponding to the annealed marginals $p_t(\mathbf{x}) \propto q_t(\mathbf{x})^\lambda$ can be performed by simulating the following weighted SDE*

$$
\begin{aligned}
d\mathbf{x}_t &= \left[ -f_t(\mathbf{x}_t) + \sigma_t^2 \lambda \nabla \log q_t(\mathbf{x}_t) \right] dt + \sigma_t d\overline{\mathbf{w}}_t \\
dw_t &= \left[ (\lambda - 1)\langle \nabla, f_t(\mathbf{x}) \rangle + \lambda(\lambda - 1)\frac{\sigma_t^2}{2} \| \nabla \log q_t(\mathbf{x}) \|^2 + \lambda g(\mathbf{x}) \right] dt
\end{aligned}
\tag{70}
$$

*Proof.* We follow the proofs of Skreta et al. (2025a).

We aim to find the partial derivative of the density $p_t(\mathbf{x}) = \frac{q_t(\mathbf{x})^\lambda}{\int q_t(\mathbf{x})^\lambda dx}$ over time $\frac{\partial p_t(\mathbf{x})}{\partial t}$, where

$$
\frac{\partial q_t(\mathbf{x})}{\partial t} = -\langle \nabla, q_t(\mathbf{x})(-f_t + \sigma_t^2 \nabla \log q_t(\mathbf{x})) \rangle + \frac{\sigma_t^2}{2} \Delta q_t(\mathbf{x}) + q_t(\mathbf{x}) \left[ \bar{g}_t(\mathbf{x}) \right].
$$

Then we have

$$
\frac{\partial \log q_t(\mathbf{x})}{\partial t} = \frac{1}{q_t(\mathbf{x})} \frac{\partial q_t(\mathbf{x})}{\partial t}
\tag{71}
$$

$$
= -\frac{1}{q_t(\mathbf{x})} \langle \nabla, \ q_t(\mathbf{x})(-f_t + \sigma_t^2 \nabla \log q_t(\mathbf{x})) \rangle + \frac{\sigma_t^2}{2} \frac{\Delta q_t(\mathbf{x})}{q_t(\mathbf{x})} + \bar{g}(\mathbf{x})
\tag{72}
$$

$$
= -\frac{1}{q_t(\mathbf{x})} \langle \nabla, \ q_t(\mathbf{x})(-f_t + \sigma_t^2 \nabla \log q_t(\mathbf{x})) \rangle + \frac{\sigma_t^2}{2} (\Delta \log q_t + \| \nabla \log q_t \|^2) + \bar{g}(\mathbf{x})
\tag{73}
$$

$$
\begin{aligned}
= &-\langle \nabla, -f_t + \sigma_t^2 \nabla \log q_t \rangle \ - \ \langle -f_t + \sigma_t^2 \nabla \log q_t, \nabla \log q_t \rangle \\
&+ \frac{\sigma_t^2}{2} (\Delta \log q_t + \| \nabla \log q_t \|^2) + \bar{g}(\mathbf{x})
\end{aligned}
\tag{74}
$$

$$
\begin{aligned}
= &\langle \nabla, f_t \rangle + \langle f_t, \nabla \log q_t \rangle - \sigma_t^2 \Delta \log q_t - \sigma_t^2 \| \nabla \log q_t \|^2 \\
&+ \frac{\sigma_t^2}{2} (\Delta \log q_t + \| \nabla \log q_t \|^2) + \bar{g}(\mathbf{x})
\end{aligned}
\tag{75}
$$

$$
= \langle \nabla, f_t \rangle + \langle f_t, \nabla \log q_t \rangle - \frac{\sigma_t^2}{2} (\Delta \log q_t + \| \nabla \log q_t \|^2) + \bar{g}(\mathbf{x}).
\tag{76}
$$

and can now compute

$$\frac{\partial \log p_t(\mathbf{x})}{\partial t} = \lambda \frac{\partial \log q_t(\mathbf{x})}{\partial t} - \int \lambda \, p_t(\mathbf{x}) \frac{\partial \log q_t(\mathbf{x})}{\partial t} \, d\mathbf{x} \tag{77}$$

$$= \lambda \left[ \langle \nabla, f_t \rangle + \langle f_t, \nabla \log q_t \rangle - \frac{\sigma_t^2}{2} \left( \Delta \log q_t + \|\nabla \log q_t\|^2 \right) + \bar{g} \right] - \int \lambda \, p_t(\mathbf{x}) \frac{\partial \log q_t(\mathbf{x})}{\partial t} \, d\mathbf{x} \tag{78}$$

$$= \lambda \langle \nabla, f_t \rangle + \lambda \langle f_t, \nabla \log q_t \rangle - \frac{\lambda \sigma_t^2}{2} \left( \Delta \log q_t + \|\nabla \log q_t\|^2 \right) + \lambda \bar{g} - \int \lambda \, p_t(\mathbf{x}) \frac{\partial \log q_t(\mathbf{x})}{\partial t} \, d\mathbf{x} \tag{79}$$

$$= \langle \nabla, \lambda f_t \rangle + \langle f_t, \nabla \log p_t \rangle - \frac{\lambda \sigma_t^2}{2} \left( \Delta \log q_t + \|\nabla \log q_t\|^2 \right) + \lambda \bar{g} - \int \lambda \, p_t(\mathbf{x}) \frac{\partial \log q_t(\mathbf{x})}{\partial t} \, d\mathbf{x} \tag{80}$$

$$= \langle \nabla, f_t \rangle + \langle f_t, \nabla \log p_t \rangle - (1-\lambda)\langle \nabla, f_t \rangle - \frac{\lambda \sigma_t^2}{2} \left( \Delta \log q_t + \|\nabla \log q_t\|^2 \right) + \lambda \bar{g} - \int p_t \left[ \langle \nabla, f_t \rangle + \langle f_t, \nabla \log p_t \rangle - (1-\lambda)\langle \nabla, f_t \rangle - \frac{\lambda \sigma_t^2}{2} \left( \Delta \log q_t + \|\nabla \log q_t\|^2 \right) + \lambda \bar{g} \right] d\mathbf{x} \tag{81}$$

$$= \langle \nabla, f_t \rangle + \langle f_t, \nabla \log p_t \rangle - (1-\lambda)\langle \nabla, f_t \rangle - \frac{\lambda \sigma_t^2}{2} \left( \Delta \log q_t + \|\nabla \log q_t\|^2 \right) + \lambda g - \int p_t \left[ -(1-\lambda)\langle \nabla, f_t \rangle - \frac{\lambda \sigma_t^2}{2} \left( \Delta \log q_t + \|\nabla \log q_t\|^2 \right) + \lambda g \right] d\mathbf{x} \tag{82}$$

$$= \langle \nabla, f_t \rangle + \langle f_t, \nabla \log p_t \rangle - (1-\lambda)\langle \nabla, f_t \rangle - \frac{\sigma_t^2}{2} \Delta \log p_t - \frac{\sigma_t^2}{2\lambda} \|\nabla \log p_t\|^2 + \lambda g - \int p_t \left[ -(1-\lambda)\langle \nabla, f_t \rangle - \frac{\sigma_t^2}{2} \Delta \log p_t - \frac{\sigma_t^2}{2\lambda} \|\nabla \log p_t\|^2 + \lambda g \right] d\mathbf{x} \tag{83}$$

$$= \langle \nabla, f_t \rangle + \langle f_t, \nabla \log p_t \rangle - (1-\lambda)\langle \nabla, f_t \rangle - \frac{\sigma_t^2}{2} \Delta \log p_t - \frac{\sigma_t^2}{2} \|\nabla \log p_t\|^2 + \left( 1 - \frac{1}{\lambda} \right) \frac{\sigma_t^2}{2} \|\nabla \log p_t\|^2 + \lambda g - \int p_t \left[ -(1-\lambda)\langle \nabla, f_t \rangle - \frac{\sigma_t^2}{2} \Delta \log p_t - \frac{\sigma_t^2}{2} \|\nabla \log p_t\|^2 + \left( 1 - \frac{1}{\lambda} \right) \frac{\sigma_t^2}{2} \|\nabla \log p_t\|^2 + \lambda g \right] d\mathbf{x} \tag{84}$$

$$= \langle \nabla, f_t \rangle + \langle f_t, \nabla \log p_t \rangle - (1-\lambda)\langle \nabla, f_t \rangle - \frac{\sigma_t^2}{2} \Delta \log p_t - \frac{\sigma_t^2}{2} \|\nabla \log p_t\|^2 + \left( 1 - \frac{1}{\lambda} \right) \frac{\sigma_t^2}{2} \|\nabla \log p_t\|^2 + \lambda g - \int p_t \left[ -(1-\lambda)\langle \nabla, f_t \rangle + \left( 1 - \frac{1}{\lambda} \right) \frac{\sigma_t^2}{2} \|\nabla \log p_t\|^2 + \lambda g \right] d\mathbf{x}. \tag{85}$$

With this, defining $g' = -(1-\lambda)\langle \nabla, f_t \rangle + (1 - \frac{1}{\lambda}) \frac{\sigma_t^2}{2} \|\nabla \log p_t\|^2 + \lambda g$ we finally have

$$\frac{\partial \log p_t}{\partial t} = \langle \nabla, f_t \rangle + \langle f_t, \nabla \log p_t \rangle - \frac{\sigma_t^2}{2} \Delta \log p_t - \frac{\sigma_t^2}{2} \|\nabla \log p_t\|^2 + g' - \int p_t(\mathbf{x}) g' d\mathbf{x} \tag{86}$$

$$\frac{\partial p_t}{\partial t} = p_t \frac{\partial \log p_t}{\partial t} \tag{87}$$

$$= p_t \left[ \langle \nabla, f_t \rangle + \langle f_t, \nabla \log p_t \rangle - \frac{\sigma_t^2}{2} \Delta \log p_t - \frac{\sigma_t^2}{2} \|\nabla \log p_t\|^2 + g'(\mathbf{x}) - \mathbb{E}_{p_t} g'(\mathbf{x}) \right] \tag{88}$$

$$= -\langle \nabla, -f_t p_t \rangle + p_t \left[ -\frac{\sigma_t^2}{2} \Delta \log p_t - \frac{\sigma_t^2}{2} \|\nabla \log p_t\|^2 + g'(\mathbf{x}) - \mathbb{E}_{p_t} g'(\mathbf{x}) \right] \tag{89}$$

$$= -\langle \nabla, -f_t p_t \rangle + p_t \left[ -\frac{\sigma_t^2}{2} \frac{\Delta p_t}{p_t} + \frac{\sigma_t^2}{2} \|\nabla \log p_t\|^2 - \frac{\sigma_t^2}{2} \|\nabla \log p_t\|^2 + g'(\mathbf{x}) - \mathbb{E}_{p_t} g'(\mathbf{x}) \right] \tag{90}$$

$$= -\langle \nabla, p_t(-f_t + \sigma_t^2 \nabla \log p_t) \rangle + \frac{\sigma_t^2}{2} \Delta p_t + p_t \left[ g'(\mathbf{x}) - \mathbb{E}_{p_t} g'(\mathbf{x}) \right] \tag{91}$$

And finally, we can reexpress this as

$$\frac{\partial p_t}{\partial t} = -\langle \nabla, p_t(-f_t + \sigma_t^2 \lambda \nabla \log q_t) \rangle + \frac{\sigma_t^2}{2} \Delta p_t + p_t \left[ g'(\mathbf{x}) - \mathbb{E}_{p_t} g'(\mathbf{x}) \right] \tag{92}$$

And for $\lambda > 0$ we can simulate this as

$$d\mathbf{x}_t = \left[ -f_t(\mathbf{x}_t) + \sigma_t^2 \lambda \nabla \log q_t(\mathbf{x}_t) \right] dt + \sigma_t d\overline{\mathbf{w}}_t$$
$$dw_t = g_t'(\mathbf{x})dt = \left[ -(1-\lambda)\langle \nabla, f_t(\mathbf{x}) \rangle + \lambda(\lambda-1)\frac{\sigma_t^2}{2} \|\nabla \log q_t\|^2 + \lambda g \right] dt \tag{93}$$

$\square$

**Proposition 6.1** (Referenced Negation as CFG+FKC, Skreta et al., 2025a)**.** *Consider two diffusion models* $q_t^1(\mathbf{x}), q_t^2(\mathbf{x})$ *defined via the Feynman-Kac equation in Equation* (6) *with weights* $g_t^i$. *The weighted SDE corresponding to the referenced negation of* $p_t \propto \neg_{q_t^2} q_t^1$ *is, with* $dw_t(\mathbf{x}) = g_t(\mathbf{x})dt$

$$d\mathbf{x}_t = \left[ -f_t(\mathbf{x}_t) + \sigma_t^2(2\nabla \log q_t^2(\mathbf{x}_t) - \nabla \log q_t^1(\mathbf{x}_t)) \right] dt + \sigma_t d\overline{\mathbf{w}}_t$$
$$g_t(\mathbf{x}) = \sigma_t^2 \|\nabla \log q_t^1(\mathbf{x}_t) - \nabla \log q_t^2(\mathbf{x}_t)\|^2 + 2g_t^2(\mathbf{x}) - g_t^1(\mathbf{x}), \tag{13}$$

*Proof.* We start with the annealed distribution $q_t^2(\mathbf{x})^2$ and the annealed pseudo-distribution $q_t^1(\mathbf{x})^{-1}$. We now try to find

$$\frac{\partial \log p_t}{\partial t} = 2\frac{\partial \log q_t^2}{\partial t} - \frac{\partial \log q_t^1}{\partial t} - \int p_t \left[ 2\frac{\partial \log q_t^2}{\partial t} - \frac{\partial \log q_t^1}{\partial t} \right] \tag{94}$$

$$= 2\frac{\partial \log q_t^2}{\partial t} - \frac{\partial \log q_t^1}{\partial t} - \int p_t \left[ 2\frac{\partial \log q_t^2}{\partial t} - \frac{\partial \log q_t^1}{\partial t} \right] \tag{95}$$

$$= 2\left[ \langle \nabla, f_t \rangle + \langle f_t, \nabla \log q_t^2 \rangle - \frac{\sigma_t^2}{2}\left( \Delta \log q_t^2 + \|\nabla \log q_t^2\|^2 \right) + \bar{g}^2(\mathbf{x}) \right] -$$
$$\left[ \langle \nabla, f_t \rangle + \langle f_t, \nabla \log q_t^1 \rangle - \frac{\sigma_t^2}{2}\left( \Delta \log q_t^1 + \|\nabla \log q_t^1\|^2 \right) + \bar{g}^1(\mathbf{x}) \right] - \tag{96}$$
$$\int p_t(\mathbf{x}) \left[ 2\frac{\partial \log q_t^2(\mathbf{x})}{\partial t} - \frac{\partial \log q_t^1(\mathbf{x})}{\partial t} \right] d\mathbf{x}$$

$$= \langle \nabla, f_t \rangle + \langle f_t, 2\nabla \log q_t^2 \rangle - \langle f_t, \nabla \log q_t^1 \rangle + 2\left[ -\frac{\sigma_t^2}{2}\left( \Delta \log q_t^2 + \|\nabla \log q_t^2\|^2 \right) + \bar{g}^2(\mathbf{x}) \right] -$$
$$\left[ -\frac{\sigma_t^2}{2}\left( \Delta \log q_t^1 + \|\nabla \log q_t^1\|^2 \right) + \bar{g}^1(\mathbf{x}) \right] - \int p_t(\mathbf{x}) \left[ 2\frac{\partial \log q_t^2(\mathbf{x})}{\partial t} - \frac{\partial \log q_t^1(\mathbf{x})}{\partial t} \right] d\mathbf{x} \tag{97}$$

$$= \langle \nabla, f_t \rangle + \langle f_t, \nabla \log p_t \rangle - \frac{\sigma_t^2}{2}\left( 2(\Delta \log q_t^2 + \|\nabla \log q_t^2\|^2) - (\Delta \log q_t^1 + \|\nabla \log q_t^1\|^2) \right) +$$
$$2\bar{g}^2(\mathbf{x}) - \bar{g}^1(\mathbf{x}) - \int p_t(\mathbf{x}) \left[ 2\frac{\partial \log q_t^2(\mathbf{x})}{\partial t} - \frac{\partial \log q_t^1(\mathbf{x})}{\partial t} \right] d\mathbf{x} \tag{98}$$

$$= \langle \nabla, f_t \rangle + \langle f_t, \nabla \log p_t \rangle - \frac{\sigma_t^2}{2}\left( \Delta \log p_t + \|\nabla \log p_t\|^2 - 2\|\nabla \log q_t^2 - \nabla \log q_t^1\|^2 \right) +$$
$$2\bar{g}^2(\mathbf{x}) - \bar{g}^1(\mathbf{x}) - \int p_t(\mathbf{x}) \left[ 2\frac{\partial \log q_t^2(\mathbf{x})}{\partial t} - \frac{\partial \log q_t^1(\mathbf{x})}{\partial t} \right] d\mathbf{x} \tag{99}$$

$$= \langle \nabla, f_t \rangle + \langle f_t, \nabla \log p_t \rangle - \frac{\sigma_t^2}{2}\left( \Delta \log p_t + \|\nabla \log p_t\|^2 \right) + \sigma_t^2 \|\nabla \log q_t^2 - \nabla \log q_t^1\|^2 +$$
$$2g^2(\mathbf{x}) - g^1(\mathbf{x}) - \mathbb{E}_{p_t}\left[ \sigma_t^2 \|\nabla \log q_t^2 - \nabla \log q_t^1\|^2 + 2g^2(\mathbf{x}) - g^1(\mathbf{x}) \right] \tag{100}$$

And with $g(\mathbf{x}) = \sigma_t^2 \|\nabla \log q_t^2 - \nabla \log q_t^1\|^2 + 2g^2(\mathbf{x}) - g^1(\mathbf{x})$

$$\frac{\partial p_t}{\partial t} = p_t \frac{\partial \log p_t}{\partial t} \tag{101}$$

$$= p_t \left[ \langle \nabla, f_t \rangle + \langle f_t, \nabla \log p_t \rangle - \frac{\sigma_t^2}{2}\left( \Delta \log p_t + \|\nabla \log p_t\|^2 \right) \right] + p_t \left[ g(\mathbf{x}) - \mathbb{E}_{p_t} g(\mathbf{x}) \right] \tag{102}$$

$$= -\langle \nabla, p_t(\mathbf{x})(-f_t + \sigma_t^2 \nabla \log p_t) \rangle + \frac{\sigma_t^2}{2}\Delta p_t + p_t \left[ g(\mathbf{x}) - \mathbb{E}_{p_t} g(\mathbf{x}) \right], \tag{103}$$

which we can simulate with

$$d\mathbf{x}_t = \left[ -f_t(\mathbf{x}_t) + \sigma_t^2(2\nabla \log q_t^2(\mathbf{x}_t) - \nabla \log q_t^1(\mathbf{x}_t)) \right] dt + \sigma_t d\overline{\mathbf{w}}_t$$
$$g_t(\mathbf{x}) = \sigma_t^2 \|\nabla \log q_t^1(\mathbf{x}_t) - \nabla \log q_t^2(\mathbf{x}_t)\|^2 + 2g_t^2(\mathbf{x}) - g_t^1(\mathbf{x}). \tag{104}$$

$\square$

**Theorem 6.2.** *Consider two weighted diffusion models $q_t^1(\mathbf{x}), q_t^2(\mathbf{x})$ defined via the Feynman-Kac equation with weights $g_t^1(\mathbf{x}), g_t^2(\mathbf{x})$, and a parameter $\lambda \in \mathbb{R}\backslash\{0\}$. The weighted SDE corresponding to $p_t(\mathbf{x}) \propto \left( q_t^1(\mathbf{x})^\lambda + q_t^2(\mathbf{x})^\lambda \right)^{1/\lambda}$, with $\alpha_t^i = \frac{q_t^i(\mathbf{x})^\lambda}{q_t^1(\mathbf{x})^\lambda + q_t^2(\mathbf{x})^\lambda} \in (0,1)$, and $dw_t = \overline{g}_t(\mathbf{x})dt$ is*

$$d\mathbf{x}_t = \left[ -f_t(\mathbf{x}_t) + \sigma_t^2(\alpha_t^1 \nabla \log q_t^1(\mathbf{x}_t) + \alpha_t^2 \nabla \log q_t^2(\mathbf{x}_t)) \right] dt + \sigma_t d\overline{\mathbf{w}}_t$$

$$g_t(\mathbf{x}) = (1 - \lambda)\frac{\sigma_t^2}{2}\left[ \left\| \sum_{i\in\{1,2\}} \alpha_t^i \nabla \log q_t^i(\mathbf{x}_t) \right\|^2 - \sum_{i\in\{1,2\}} \alpha_t^i \|\nabla \log q_t^i(\mathbf{x}_t)\|^2 \right] + \sum_{i\in\{1,2\}} \alpha_t^i g_t^i(\mathbf{x}_t). \tag{14}$$

*Proof of Theorem 6.2.* We now use our two lemmas to show the main result. We begin with

$$d\mathbf{x}_t = \left[-f_t(\mathbf{x}_t) + \sigma_t^2 \lambda \nabla \log q_t^i(\mathbf{x}_t)\right] dt + \sigma_t d\overline{\mathbf{w}}_t$$
$$dw_t = (\lambda - 1) \left(\langle \nabla, f_t(\mathbf{x}_t)\rangle + \frac{\sigma^2}{2}\lambda \|\nabla \log q_t^i(\mathbf{x}_t)\|^2\right) dt + \lambda g_t^i(\mathbf{x}) \tag{105}$$

for both annealed distributions, according to Lemma C.2. Then, by Lemma C.1, we have a mixture of these distributions with

$$d\mathbf{x}_t = \left[-f_t(\mathbf{x}_t) + \sigma_t^2 \lambda (\alpha_t^1 \nabla \log q_t^1(\mathbf{x}_t) + \alpha_t^2 \nabla \log q_t^2(\mathbf{x}_t))\right] dt + \sigma_t d\overline{\mathbf{w}}_t$$
$$dw_t = \alpha_t^1 \left[(\lambda - 1)\left(\langle \nabla, f_t(\mathbf{x}_t)\rangle + \frac{\sigma^2}{2}\lambda \|\nabla \log q_t^1(\mathbf{x}_t)\|^2\right) dt + \lambda g_t^1(\mathbf{x})\right] +$$
$$\alpha_t^2 \left[(\lambda - 1)\left(\langle \nabla, f_t(\mathbf{x}_t)\rangle + \frac{\sigma^2}{2}\lambda \|\nabla \log q_t^2(\mathbf{x}_t)\|^2\right) dt + \lambda g_t^2(\mathbf{x})\right] \tag{106}$$

which simplifies to

$$dw_t = (\lambda - 1)\langle \nabla, f_t(\mathbf{x}_t)\rangle dt + \lambda \left[\sum_{i \in \{1,2\}} \alpha_t^i \left((\lambda - 1)\frac{\sigma^2}{2}\|\nabla \log q_t^i(\mathbf{x}_t)\|^2 dt + g_t^i(\mathbf{x}_t)\right)\right]. \tag{107}$$

Finally, we apply Lemma C.2 to the resulting mixture with $1/\lambda$. This then results in

$$d\mathbf{x}_t = \left[-f_t(\mathbf{x}_t) + \sigma_t^2 (\alpha_t^1 \nabla \log q_t^1(\mathbf{x}_t) + \alpha_t^2 \nabla \log q_t^2(\mathbf{x}_t))\right] dt + \sigma_t d\overline{\mathbf{w}}_t, \tag{108}$$

which is the target score as desired. For our weight-field we then have

$$dw_t = \begin{aligned}&(\frac{1}{\lambda} - 1)\left(\langle \nabla, f_t(\mathbf{x}_t)\rangle + \frac{\sigma^2}{2}\frac{1}{\lambda}\|\alpha_t^1 \lambda \nabla \log q_t^1(\mathbf{x}_t) + \alpha_t^2 \lambda \nabla \log q_t^2(\mathbf{x}_t)\|^2\right) dt+ \\ &\frac{1}{\lambda}\left[(\lambda - 1)\langle \nabla, f_t(\mathbf{x}_t)\rangle + \lambda \left[\sum_{i \in \{1,2\}} \alpha_t^i \left((\lambda - 1)\frac{\sigma^2}{2}\|\nabla \log q_t^i(\mathbf{x}_t)\|^2 + g_t^i(\mathbf{x}_t)\right)\right]\right] dt\end{aligned} \tag{109}$$

$$= \begin{aligned}&\frac{1 - \lambda}{\lambda}\langle \nabla, f_t(\mathbf{x}_t)\rangle dt + \frac{1 - \lambda}{\lambda}\frac{\sigma^2}{2}\frac{1}{\lambda}\|\alpha_t^1 \lambda \nabla \log q_t^1(\mathbf{x}_t) + \alpha_t^2 \lambda \nabla \log q_t^2(\mathbf{x}_t)\|^2 dt+ \\ &\frac{\lambda - 1}{\lambda}\langle \nabla, f_t(\mathbf{x}_t)\rangle dt + \sum_{i \in \{1,2\}} \alpha_t^i \left((\lambda - 1)\frac{\sigma^2}{2}\|\nabla \log q_t^i(\mathbf{x}_t)\|^2 + g_t^i(\mathbf{x}_t)\right) dt\end{aligned} \tag{110}$$

$$= \begin{aligned}&(1 - \lambda)\frac{\sigma^2}{2}\left\|\sum_{i \in \{1,2\}} \alpha_t^i \nabla \log q_t^i(\mathbf{x}_t)\right\|^2 dt+ \\ &\sum_{i \in \{1,2\}} \alpha_t^i \left((\lambda - 1)\frac{\sigma^2}{2}\|\nabla \log q_t^i(\mathbf{x}_t)\|^2 + g_t^i(\mathbf{x}_t)\right) dt\end{aligned} \tag{111}$$

$$= (1 - \lambda)\frac{\sigma^2}{2}\left[\left\|\sum_{i \in \{1,2\}} \alpha_t^i \nabla \log q_t^i(\mathbf{x}_t)\right\|^2 - \sum_{i \in \{1,2\}} \alpha_t^i \|\nabla \log q_t^i(\mathbf{x}_t)\|^2\right] dt + \sum_{i \in \{1,2\}} \alpha_t^i g_t^i(\mathbf{x}_t) dt \tag{112}$$

We can see that, as expected, for $\lambda = 1$ we are left with the unweighted mixture of distributions. For more complex compositions, the weight fields just propagate as well, we can see that the statement trivially generalizes to more than two diffusion models, so we maintain associativity.

$$\square$$

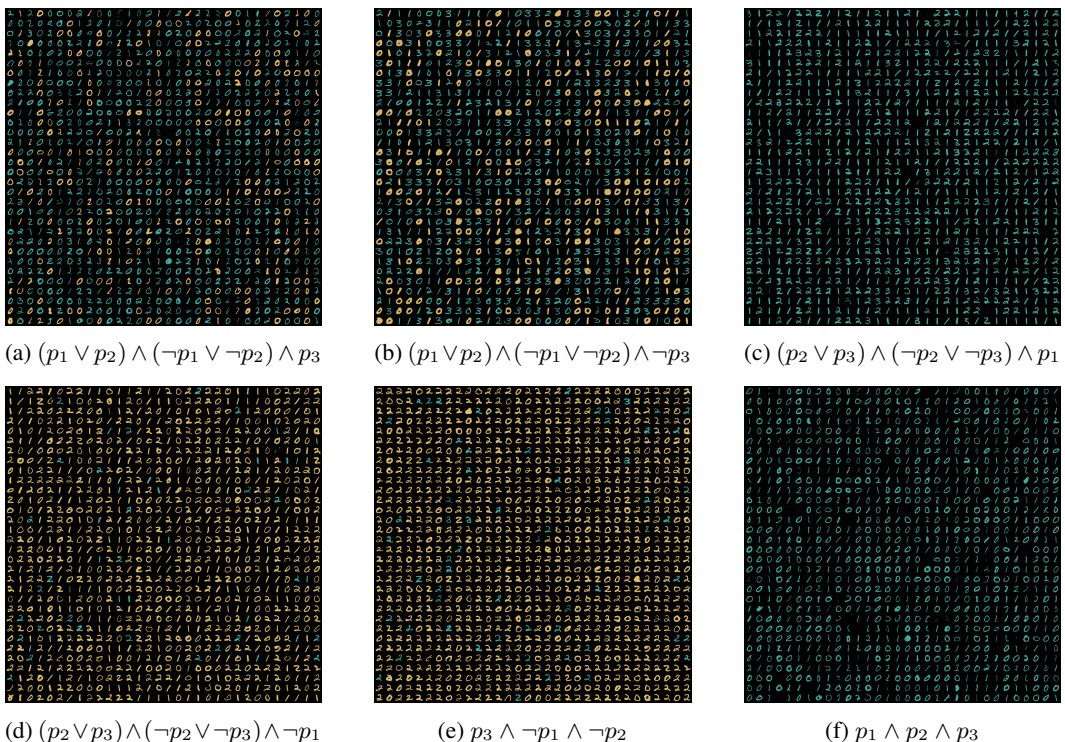

(a) $(p_1 \vee p_2) \wedge (\neg p_1 \vee \neg p_2) \wedge p_3$  (b) $(p_1 \vee p_2) \wedge (\neg p_1 \vee \neg p_2) \wedge \neg p_3$  (c) $(p_2 \vee p_3) \wedge (\neg p_2 \vee \neg p_3) \wedge p_1$

(d) $(p_2 \vee p_3) \wedge (\neg p_2 \vee \neg p_3) \wedge \neg p_1$  (e) $p_3 \wedge \neg p_1 \wedge \neg p_2$  (f) $p_1 \wedge p_2 \wedge p_3$

Figure 7: Generated MNIST score compositions.

## D    EXPERIMENTS

All our experiments on stable diffusion and SBDD were performed on unmodified, pretrained models. We performed inference on Nvidia V100 and A100 GPUs.

### D.1    MNIST EXPERIMENTS

We reproduce the setup of (Garipov et al., 2023), and generate images from the score composition of the three toy MNIST models. The code to train the models is available in the code repository, and training was completed on a Nvidia GTX 3080 desktop in 10 minutes.

We show image collages for non-trivial example formulas in Figure 7. For each formula, we generated a batch of 1024 images.

### D.2    STABLE DIFFUSION IMAGE GENERATION

We reproduce the stable diffusion experimental setup of (Skreta et al., 2025b) with Stable Diffusion v1-4 available pretrained publicly at huggingface: https://huggingface.co/CompVis/stable-diffusion-v1-4. We then report PoE, SuperDiff's and as well as joint prompts.

We use 20 pairs of conjunctive prompt pairs and generate 20 images each. We provide a batch of the generated images in the supplementary material and list the prompts here, also reused from (Skreta et al., 2025b):

- "a mountain landscape" ∧ "silhouette of a dog"
- "a flamingo" ∧ "a candy cane"
- "a dragonfly" ∧ "a helicopter"
- "dandelion" ∧ "fireworks"

- `"a sunflower"` ∧ `"a lemon"`
- `"a rocket"` ∧ `"a cactus"`
- `"moon"` ∧ `"cookie"`
- `"a snail"` ∧ `"a cinnamon roll"`
- `"an eagle"` ∧ `"an airplane"`
- `"zebra"` ∧ `"barcode"`
- `"chess pawn"` ∧ `"bottle cap"`
- `"a pineapple"` ∧ `"a beehive"`
- `"a spider web"` ∧ `"a bicycle wheel"`
- `"a waffle cone"` ∧ `"a volcano"`
- `"a cat"` ∧ `"a dog"`
- `"a chair"` ∧ `"an avocado"`
- `"a donut"` ∧ `"a map"`
- `"otter"` ∧ `"duck"`
- `"pebbles on a beach"` ∧ `"a turtle"`
- `"teddy bear"` ∧ `"panda"`

For the contrastive Prompts, we partially use our own prompts and partially use the prompts from (Dong et al., 2023). We provide a batch of the generated images in the supplementary material. and list the prompts here:

- `"A night sky with stars and a crescent moon, reminiscent of Van Gogh's 'Starry Night'."` ∧¬ `"Van Gogh"`
- `"A night sky with stars and a crescent moon, reminiscent of Van Gogh's 'Starry Night'."` ∧¬ `"Picasso's Cubist style"`
- `"A portrait of a man with a distorted and fragmented face painted in Picasso's Cubist style."` ∧¬ `"Picasso's Cubist style"`
- `"A cat and a ball on the shelf"` ∧¬ `"cat, ball"`
- `"There are a bicycle and a car in front of the house"` ∧¬ `"a bicycle and a car"`
- `"orange fruit"` ∧¬ `"orange color palette"`
- `"a banana"` ∧¬ `"yellow color palette"`
- `"an ocean"` ∧¬ `"blue color palette"`
- `"strawberry"` ∧¬ `"red color palette"`
- `"round shape"` ∧¬ `"circle"`

### D.2.1 ADDITIONAL RESULTS

We provide additional plots illustrating the behavior of composition under varying values of $\lambda$ in Figure 8.

### D.3 ADDITIONAL RESULTS ON SBDD MOLECULE GENERATION

We report a sweep across three values of $\lambda$ for the molecule generation task in Table 3. As the variance in this experiment is high, none of the differences can be considered significant.

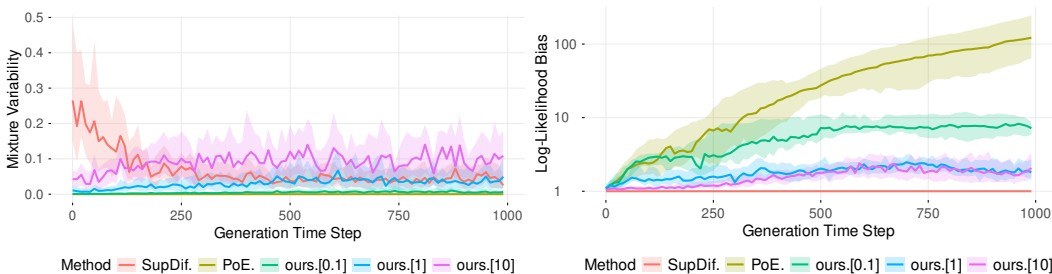

(a) Variability of mixture coefficients for conjunction  (b) Absolute difference in likelihood during generation

Figure 8: Mixture Stability vs Likelihood Bias in SD experiment. Figure 8a shows the absolute change rate of $\alpha^i$, Figure 8b shows the median absolute log-density ratio. PoE (or geometric mean) has constant mixture coefficients, but log-likelihoods diverge during the diffusion process. Superdiff forces equal likelihoods as the cost of a highly variable mixture, especially early during the diffusion process. Dombi composition (ours.[$\lambda$]) provides a tradeoff, depending on $\lambda$.

Table 3: Docking Scores of generated ligands for 14 protein target pairs ($P_1$, $P_2$), in batches of 32 ligands for 5 molecule lengths each. Extended runs across temperatures $\gamma \in \{1, 2\}$. We compare conjunction with Dombi with various $\lambda$ with and without FKC with annealed base distribution and also report TargetDiff from (Guan et al., 2023) as baseline.

| Method | Temp. $\gamma$ | $\lambda$ | FKC? | ($P_1 * P_2$) ($\uparrow$) | max($P_1$, $P_2$) ($\downarrow$) | Better than ref. ($\uparrow$) | Div. ($\uparrow$) | Val. & Uniq. ($\uparrow$) | QED ($\uparrow$) | SA ($\downarrow$) |
|---|---|---|---|---|---|---|---|---|---|---|
| TargetDiff | – | – | – | $62.19_{\pm 27.08}$ | $-7.24_{\pm 2.35}$ | $0.32_{\pm 0.37}$ | $\mathbf{0.89}_{\pm 0.01}$ | $0.95_{\pm 0.07}$ | $0.57_{\pm 0.14}$ | $\mathbf{0.59}_{\pm 0.09}$ |
| Dombi | 1 | 0.3 | ✗ | $68.12_{\pm 27.38}$ | $-7.37_{\pm 2.51}$ | $0.26_{\pm 0.32}$ | $0.88_{\pm 0.02}$ | $\mathbf{0.96}_{\pm 0.10}$ | $0.58_{\pm 0.12}$ | $\mathbf{0.59}_{\pm 0.10}$ |
| Dombi | 1 | 1 | ✗ | $68.60_{\pm 28.09}$ | $-7.42_{\pm 2.57}$ | $0.28_{\pm 0.34}$ | $0.88_{\pm 0.02}$ | $\mathbf{0.96}_{\pm 0.09}$ | $0.58_{\pm 0.13}$ | $\mathbf{0.59}_{\pm 0.10}$ |
| Dombi | 1 | 3 | ✗ | $67.92_{\pm 28.17}$ | $-7.33_{\pm 2.61}$ | $0.28_{\pm 0.34}$ | $0.88_{\pm 0.01}$ | $\mathbf{0.96}_{\pm 0.09}$ | $0.57_{\pm 0.13}$ | $\mathbf{0.59}_{\pm 0.10}$ |
| Dombi | 1 | 0.3 | ✓ | $72.09_{\pm 31.16}$ | $-7.51_{\pm 2.64}$ | $0.31_{\pm 0.37}$ | $0.87_{\pm 0.02}$ | $0.95_{\pm 0.12}$ | $0.56_{\pm 0.13}$ | $\mathbf{0.59}_{\pm 0.11}$ |
| Dombi | 1 | 1 | ✓ | $72.83_{\pm 22.42}$ | $-7.71_{\pm 1.65}$ | $0.27_{\pm 0.35}$ | $0.86_{\pm 0.03}$ | $0.95_{\pm 0.08}$ | $0.57_{\pm 0.13}$ | $\mathbf{0.59}_{\pm 0.11}$ |
| Dombi | 1 | 3 | ✓ | $70.01_{\pm 27.94}$ | $-7.50_{\pm 2.50}$ | $0.28_{\pm 0.33}$ | $0.86_{\pm 0.02}$ | $\mathbf{0.96}_{\pm 0.10}$ | $0.58_{\pm 0.13}$ | $0.61_{\pm 0.09}$ |
| Dombi | 2 | 0.3 | ✗ | $72.54_{\pm 29.03}$ | $-7.67_{\pm 2.41}$ | $0.32_{\pm 0.35}$ | $0.88_{\pm 0.02}$ | $0.93_{\pm 0.16}$ | $0.59_{\pm 0.13}$ | $0.61_{\pm 0.10}$ |
| Dombi | 2 | 1 | ✗ | $71.36_{\pm 29.44}$ | $-7.59_{\pm 2.48}$ | $0.30_{\pm 0.34}$ | $0.88_{\pm 0.01}$ | $0.93_{\pm 0.16}$ | $0.59_{\pm 0.12}$ | $0.62_{\pm 0.09}$ |
| Dombi | 2 | 3 | ✗ | $72.92_{\pm 29.50}$ | $-7.74_{\pm 2.46}$ | $0.31_{\pm 0.36}$ | $0.88_{\pm 0.02}$ | $0.94_{\pm 0.16}$ | $\mathbf{0.60}_{\pm 0.12}$ | $0.62_{\pm 0.09}$ |
| Dombi | 2 | 0.3 | ✓ | $78.75_{\pm 33.36}$ | $-7.98_{\pm 2.51}$ | $0.37_{\pm 0.40}$ | $0.87_{\pm 0.03}$ | $0.94_{\pm 0.15}$ | $0.59_{\pm 0.12}$ | $0.61_{\pm 0.10}$ |
| Dombi | 2 | 1 | ✓ | $81.63_{\pm 25.91}$ | $-8.25_{\pm 1.56}$ | $0.38_{\pm 0.40}$ | $0.85_{\pm 0.11}$ | $0.93_{\pm 0.17}$ | $0.59_{\pm 0.12}$ | $0.62_{\pm 0.10}$ |
| Dombi | 2 | 3 | ✓ | $\mathbf{83.06}_{\pm 27.02}$ | $-\mathbf{8.40}_{\pm 1.61}$ | $\mathbf{0.40}_{\pm 0.41}$ | $0.85_{\pm 0.03}$ | $0.94_{\pm 0.12}$ | $0.57_{\pm 0.13}$ | $0.62_{\pm 0.09}$ |

## E  EXPERIMENT FOR COMBINATORIAL CONSTRAINTS

### E.1  SETUP

We illustrate the capability of Dombi compositions to adhere to combinatorial constraints by sampling uniformly from satisfying variable assignments of propositional formulas. For a formula with $k$ propositional variables $P_i$, for $i \in [1, k]$, we set up our diffusion ensemble as follows: In $\mathbb{R}^k$, we place $2^k$ Gaussian modes, one for each possible variable assignment. Then, in our ensemble, each of $k$ score models simulates one propositional variable. For $i \in [1, k]$, we have access to $s_i$, which defines a denoising process to a uniform mixture of the $2^{k-1}$ Gaussian modes, where the $P_i$ is true. Additionally, a reference model defines a denoising process uniformly to *all* $2^k$ Gaussian modes. For $k = 2$, this setup is visualized in Figure 9a.

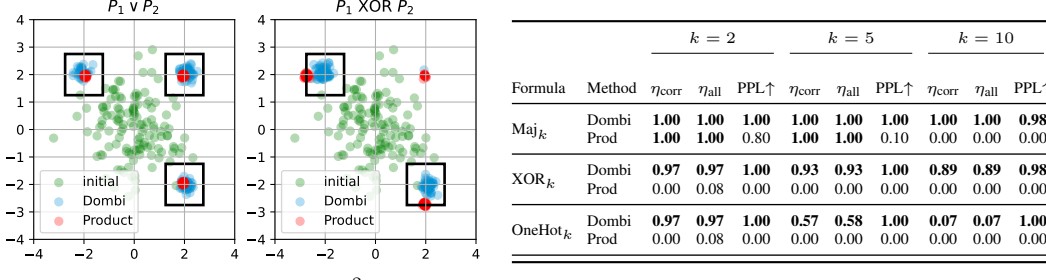

(a) SAT experiment in $\mathbb{R}^2$.

(b) Overview of SAT experiment for three formulas.

|  |  | $k = 2$ | | | $k = 5$ | | | $k = 10$ | | |
|---|---|---|---|---|---|---|---|---|---|---|
| Formula | Method | $\eta_{\text{corr}}$ | $\eta_{\text{all}}$ | PPL↑ | $\eta_{\text{corr}}$ | $\eta_{\text{all}}$ | PPL↑ | $\eta_{\text{corr}}$ | $\eta_{\text{all}}$ | PPL↑ |
| $\text{Maj}_k$ | Dombi | **1.00** | **1.00** | **1.00** | **1.00** | **1.00** | **1.00** | **1.00** | **1.00** | **0.98** |
|  | Prod | **1.00** | **1.00** | 0.80 | **1.00** | **1.00** | 0.10 | 0.00 | 0.00 | 0.00 |
| $\text{XOR}_k$ | Dombi | **0.97** | **0.97** | **1.00** | **0.93** | **0.93** | **1.00** | **0.89** | **0.89** | **0.98** |
|  | Prod | 0.00 | 0.08 | 0.00 | 0.00 | 0.00 | 0.00 | 0.00 | 0.00 | 0.00 |
| $\text{OneHot}_k$ | Dombi | **0.97** | **0.97** | **1.00** | **0.57** | **0.58** | **1.00** | **0.07** | **0.07** | **1.00** |
|  | Prod | 0.00 | 0.08 | 0.00 | 0.00 | 0.00 | 0.00 | 0.00 | 0.00 | 0.00 |

Figure 9: Figure 9a shows the SAT experiment in $\mathbb{R}^2$, with squares corresponding to satisfying assignments. The corresponding numerical overview for $k \in \{2, 5, 10\}$ in Figure 9b. Best are bold.

Our objective is then to use score-composition to uniformly sample from all satisfying variable assignments. We repeat this setup for the Dombi operators, as well as PoE/MoE composition for three formulas for $k \in [1, 10]$, and report mode coverage, uniformity, and stability of the composition.

### E.1.1  SAT FORMULAS

We use three different propositional formulas: majority, xor, and one-hot. The formulations of these formulas are designed to test different aspects of the score composition.

**Majority**  We define the formula over $k$ variables as

$$\text{Maj}_k(P_1, \ldots, P_k) \equiv \bigwedge_{\substack{S \subseteq \{P_1, \ldots, P_k\} \\ |S| = \lceil k/2 \rceil}} \bigvee_{P \in S} .$$

This formula is negation-free, but might lead to mode dropping for variable assignments with fewer positive variables.

**One-Hot**  We define a formula where exactly one variable has to be true as

$$\text{OneHot}_k(P_1, \ldots, P_k) \equiv \left( \bigvee_{i=1}^{k} P_i \right) \wedge \left( \bigwedge_{1 \leq i < j \leq k} (\neg P_i \vee \neg P_j) \right).$$

It is only quadratic in the length of the variables, but it contains many clauses without positive literals, requiring precise handling of explicit negation.

**Exclusive Or**  We define xor as a parity function over $k$ variables as

$$\text{XOR}_k(P_1, \ldots, P_k) \equiv \bigwedge_{\substack{v \in \{0,1\}^k \\ \sum_i v_i \equiv 0 \ (\text{mod } 2)}} \bigvee_{i=1}^{k} (v_i? \neg P_i : P_i).$$

This formula can only be expressed in exponential length with $2^{k-1}$ clauses, which explicitly exclude one assignment with even parity.

## E.2   SCORE MODEL SETUP

We translate each of the $2^k$ propositional variable assignments to a Gaussian mode in $\mathbb{R}^k$ as

$$p(\mathbf{x}) = \frac{1}{2^k} \sum_{v \in \{0,1\}^k} \mathcal{N}_k(\mathbf{x}|4v - 2, \sigma^2).$$

We then define "directional" diffusion models

$$\forall i \in [1, k] : p_i(\mathbf{x}) = \frac{1}{2^{k-1}} \sum_{\substack{v \in \{0,1\}^k \\ v_i = 1}} \mathcal{N}_k(\mathbf{x}|4v - 2, \sigma^2).$$

In this setup, each distribution plays the role of one propositional variable. The distributions $p_i$ can then be composed to mirror a propositional formula, with the goal that particles converge only to modes that correspond to satisfying variable assignments. We use $p$ as an additional stabilizing model to guide particles to any location that corresponds to an assignment.

As these models are mixtures of Gaussians, we derive optimal scores and energy functions from the standard Gaussian to our distributions in closed form.

We then model each type of formula for $k \in [1, 10]$ as direct composition and simulate $2^{14}$ particles over 100 denoising steps.

For each mode, when then check a $L_\infty$ bounding box around its mean of sidelength $3\sigma$ and consider all particles within that radius to be valid assignments.

In Figure 9b we show the most important metrics: $\eta_{\text{corr}}$, the fraction of particles within bounding boxes of satisfying modes, $\eta_{\text{all}}$, the fraction of particles converging to any mode. Additionally, we measure the normalized perplexity in the particle distributions across as PPL. In this experiment, PPL measures mode uniformity, where a higher number indicates more uniform samples from satisfying modes of the formula. In a formula with $K$ satisfying variable assignments, for a batch of $n$ particles, with $n\eta_{\text{corr}}$ particles within satisfying modes, we denote the fraction of particles within the bounding box of the *assignment index* $i \in [1, K]$ as $\eta_i$ with $\sum_i \eta_i = \eta_{\text{corr}}$. We then calculate PPL for mode confusion as

$$\text{PPL} = 2^{\left(-\sum_{i=1}^{K} \frac{\eta_i}{\eta_{\text{corr}}} \log_2 \frac{\eta_i}{\eta_{\text{corr}}}\right)} / K.$$

## E.3   RESULTS

Figure 9a shows samples of formulas in $\mathbb{R}^2$. An overview of the experimental results is provided in Figure 9b. We can see multiple shortcomings of products in our experimental results. On the negation-free $\text{Maj}_k$, PoE drastically reduces the per-mode variance, as seen in Figure 9a, drops most of the modes for $k = 5$, and completely breaks down for $k = 10$. In contrast to this, the dombi Operators do not drop modes and maintain a close-to-uniform distribution over modes in high dimensions. For $\text{XOR}_k$ and $\text{OneHot}_k$ PoE breaks down for $k = 2$ already, due to the negated literals. In Figure 9a, the modes of the PoE sample appear drastically biased by the negated clause. Somewhat surprisingly, the Dombi composition can sample comparatively well from the exponentially sized $\text{XOR}_{10}$, and struggles much more for OneHot, which is comprised of many purely negative clauses.

