# OpenReview forum: "Composition of Pretrained Diffusion Models: A Logic-Based Calculus"
_ICLR.cc/2026/Conference — ICLR 2026 Poster_

### Official Review · Reviewer_nPjQ · 2025-10-21

**Soundness:** 3
**Presentation:** 3
**Contribution:** 3
**Rating:** 6
**Confidence:** 3

**Summary:**

This paper proposes Dombi operators as a principled way of implementing conjunction, disjunction, and negation operations for composing diffusion models. The Dombi operators seek to correct flaws in the standard implementations of the composition operations including bias, instability, and failure to obey compositional laws. The proposal is tested on image and molecular generation problems.

**Strengths:**

I appreciate the rigorous derivation of "correct" operators for conjunction, disjunction, and negation. The perspective is helpful for clarifying and unifying a somewhat-messy landscape in which different works use different implementations of compositional operators loosely tied to logical operators -- but which may not represent those logical operators very well, and also may not combine together (e.g. conjunction with negation) in logically consistent ways.

**Weaknesses:**

The analysis of the flaws in PoE/MoE methods, as well as the comparison of the Dombi operators to "standard implementations", make particular choices of implementation that are in some sense the "naive" ones rather than what I would consider the "standard" ones: for example, implementing conjunction as $p^1(x) p^2(x)$ and negation as $p^1(x) / p^2(x)^\gamma$. Works (referenced in this paper but not discussed in great detail) such as Du23 and Liu22 implement conjunction not as a simple product but actually $p(x) \prod_i p(x|c_i) / p(x)$ (for two distributions, $p(x|c_1)p(x|c_2) / p(x)$), and negation not as $p(x|c_1)/p(x|c_2)$ but $p(x) p(x|c_1) / p(x|c_2)$, where $p(x)$ is the unconditional. Also, CFG is essentially a negation $p(x|c)^\gamma p(x)^{1 - \gamma} = p(x) (p(x|c)/p(x))^\gamma$. Choices like these work better in practice, and seem to be closer to the Dombi operators derived in Def 4.1. (For example the denominator in (8) is somewhat related to the unconditional). I also suspect that these implementations might not suffer from some of the failure cases (instability, failure to compose, etc.) I feel that these Du/Liu variant would be a valuable (fairer) baseline for comparison. (I still think there is theoretical value in the derivations of the Dombi forms even if they turn out to be similar to the Du/Liu variants, because to my knowledge the forms described in these earlier works were essentially discovered to work well empirically without much theoretical justification.)

**Questions:**

Can you comment on connections to the Du/Liu variants of compositional operators as discussed in Weaknesses? Did you try any of these variants for comparison, or test whether they suffer from the identified failure cases?

Discussion around Definition 4.1: I wonder if there is any connection with parametrization-independence here. Bradley&Nakkiran25 (Appendix I) (https://arxiv.org/pdf/2502.04549) (and possibly other references) suggest that parametrization-independence may be a beneficial property for compositional operators (a property possessed by the CFG operator and the Du/Liu styles of product composition); they also show that all 1-homogeneous operators are independent of parametrization. The Dombi operators you obtain appear to be 1-homogeneous as well. Can you comment on potential connections with parametrization-independence and/or other reasons that 1-homogeneity may be desirable for these operators?

Minor comments:
L123 “important to note”?
L142: CFG is usually defined as $p^1(x)^\gamma p^2(x)^(1 - gamma) = p^2(x) (p^1(x)/p^2(x))^\gamma$ — not sure what is meant here?
L215 I think this is the first time $\phi_c$ is mentioned; if so, not very well motivated/introduced
L244 Without reading Appendix A, it’s not clear what $g$ is

---

> ### Author Response · Authors · 2025-11-21
> **Review Response: Connection to Related work clarified and Different Motivations for our Negation discussed in Revision**
>
> We thank the reviewer for their kind words, especially bringing our attention to the limited discussion of the Du/Liu variant of negation operators.
> We have properly contextualised this, and are able to provide a cleaner presentation of $\phi_c$ and the negation after considering the reviewers' comments!
>
> ## Weaknesses
> **Connection to Du/Liu Negation: Discussed in depth in the revised version of our manuscript.**
> We changed the discussion of negation in related work and now differentiate between the CFG-style variant by Du and Liu $c(x)^{1+\gamma}/p(x)^\gamma$ and the ICN style negation $c(x)/p(x)^\gamma$.
> Our negation much more resembles the CFG-style negation, and is equivalent to CFG for the case $\gamma=1$.
>
> This choice is well-motivated from a perspective of probability theory and fuzzy logic, without invoking conditional generation.
> From a probabilistic perspective, we argue that bounded $\chi^2$ divergence results in normalizable negated distributions.
> Further, this choice maintains the prior during denoising without distortion.
> That is, if all operands as well as the reference are the same noise prior, their resulting composition is the same noise prior up to a constant factor.
> From the perspective of fuzzy logic, this form of negation enables scaling by a meaningful fixed-point.
>
> One specific difference, however, between the CFG-style negation and our setting is that we require our negation operation to be involutive to obtain De Morgan dual operators.
> This can only be the case for $\gamma=1$.
>
> One interesting similarity to the density ratios presented by Du and Liu can be observed when manipulating our presentation of $\phi$.
>  Our mapping function is defined as
> $\phi_{c(x)}(p(x)) = \frac{p(x)}{p(x)+c(x)}= \frac{\frac{p(x)}{c(x)}}{\frac{p(x)}{c(x)}+1}$.
> This is equivalent to normalizing $p(x)$ by $c(x)$, which is analogous to the probability ratios of Du/Liu.
> The inverse function $\phi_c^{-1}$ would then multiply the result of the composition by the unconditional again.
> This presentation of our results is mathematically cleaner than the referenced negation as we present it, and we would like to emphasize this point.
>
> We incorporated the feedback in the manuscript and will add a section to the appendix to discuss the arguments here and in our response to 9bJN.
>
> ### Identified Failure Cases.
> **Du/Liu do not address failure modes of PoE**:
> First, the issues that PoE inherently has with respect to temperature scaling are not influenced by the usage of density ratios.
> This different parametrisation means that the resulting composition is uniformly annealed *down*, as the geometric mean.
> Because the magnitude of scores might increase, stay constant, or decrease, depending on the neighborhood, annealing just shifts this instability downwards.
> This can result in an effective temperature that is too low in some regions but not others.
>
> **Unstable behaviour of negation can be prevented with Dombi conjunction:**
> As we discussed above, our negation operator also follows the CFG-style negation
> As such, the stability of this is the same as for us *in isolation*.
>
> However, in Fig. 2a, the figure illustrating unstable negation, there is an additional important aspect that is more closely related to our conjunction than the negation itself.
> *A Dombi conjunction is bounded above by the minimum.*
> Consequently, it is simply impossible to yield any unstable/non-normalizable composition under any composition of the form $(...) \land p(x)$.
> In isolation, negations are unstable if the distributions have unbounded $\chi^2$ divergence.
> PoE cannot "protect" against this instability in all scenarios.
>
> As this "failure mode" is also a problem studied for CFG ([What does guidance do?](https://doi.org/10.52202/079017-2698)), we investigate whether and to what degree our conjunction can "protect" against this in our SAT experiment.
>
> ## Questions
> **Dombi Operators are parameterisation independent.** We thank the reviewer for the resource.
> Parameter independence is a natural, desirable property: the result should not be affected by representation choices of its inputs.
> Bradley and Nakkiran state that any pointwise, 1-homogeneous operator is parameterization independent.
> This applies to Dombi operators: (pseudo) power norms are 1-homogeneous, for any $\lambda \neq 0$, and they operate strictly pointwise.
>
> **1-homogeneity in score calculus:**
> Another possible observation why 1-homogeneity is natural might be related to the fact that we perform score calculus: (constant) scaling operations on the probability distributions result in additive log-likelihood terms, which vanish for score operations.
> A failure of this property would seem to affect, at the very least, the magnitude (leading to undesired temperature scaling), or even worse, the direction of the composition under parameterization.
>
> We thank the reviewer again for their insightful arguments and additional resources! We hope for further fruitful discussion.

---

> > ### Comment · Reviewer_nPjQ · 2025-11-25
> >
> > Thank you for your responses, in particular the contextualization w.r.t. Du/Liu. I will keep my positive score.

---

### Official Review · Reviewer_ndFP · 2025-10-24

**Soundness:** 4
**Presentation:** 3
**Contribution:** 4
**Rating:** 8
**Confidence:** 3

**Summary:**

The paper examines techniques to compose diffusion models with set theoretic operations such as unions and negations. They find inconsistencies in existing techniques, in particular for negation and complex combinations of operations. They then address these problems with Dombi operators which they derive rigorously from fuzzy logic. They show that their operators are more well behaved and allows for consistent composition and algebraic operations and show more stable sampling of their method on three datasets resulting in higher sample quality.

**Strengths:**

- Composing diffusion models is extremely useful, in particular negation has many practical use cases.
- The paper clearly demonstrates the shortcomings of existing approaches to motivate the need for their method.
- The operators are rigorously derived from fuzzy logic, putting their method on firm mathematical grounds
- The experimental results convincingly demonstrate the effectiveness of their method

**Weaknesses:**

- The presentation of the paper could be slightly improved. For example, the different colors in figure 2 are too similar, making it hard to read. - - It is also not clear which of the dark curves represents which probability distribution
- Some variables seem to be undefined or fall out of nowhere, like the form of g(x) and s_c in definition 4.1
- The used lambda hyperparameter seems to very significantly between different experiments, raising the question how carefully this has to be tuned

**Questions:**

- How sensitive is the lambda hyperparameter?
- Did you try larger logical formulas? How robust is this?
- How expensive is the sampling; does the integration become more expensive in particular with the mentioned oscillations, and does the construction of the composition incur any computational overhead?

---

> ### Author Response · Authors · 2025-11-21
> **Review Response: Additional Experiment with Highly Complex Compositions, Visualisations for the behaviour of Lambda**
>
> We sincerely thank the reviewer for their very positive comments, their enthusiasm about our work, and their valuable feedback.
> We further illustrated the behaviour of $\lambda$ in our revised manuscript.
>
> ## Weaknesses
> - **Figure 2: Visual Clarity will be improved for camera-ready, with emphasis on 2.b.**
>   Thank you for bringing this to our attention. We will refine this figure in the camera-ready version of our manuscript.
>   In addition to the visual clarity and color scheme, we believe that increased emphasis should be placed on Figure 2.b, as the left and right subfigures are already elaborated upon more in the text examples.
>
>   Additionally, we included another figure (Figure 1) to illustrate our composition method in comparison to existing methods, partially inspired by reviewer nPjQ.
>   We would also greatly appreciate feedback regarding the presentation of this figure.
> - **Mathematical Clarity in definitions: Revised in updated version of the manuscript**. Thank you for pointing this out, especially $s_c$. We have revised this part of the manuscript and hope that the presentation, especially of Definition 4.1, is now clear, and all concepts are properly introduced.
>   We removed $g(x)$ (which was unused) from Definition 4.1, and defined $s_c$ as the score of the reference function $c$.
>
> ## Questions
> - **Choice of Lambda: Robust against over-/underscaling, but trades off stability and exactness.**
>   It is true that lambda varies greatly between experiments; however, it is not very sensitive.
>   Moreover (as we try to illustrate in the new Figure 1), it "fails safely".
>   We have now added **additional figures, which visualize this tradeoff in Stable Diffusion to Appendix C.2.1. (Figure 7)** and shows stable behavior.
>   If $\lambda$ is very small, the composition behaves more like PoE.
>   This still yields reasonable compositions, as PoE usually does, but does not adhere to the properties we would like to see from our logic (Corollary 5.1).
>   For large values of $\lambda$, the composition more exactly behaves like logic, but the mixing coefficients are more variable: oscillations where effectively only one component is optimized for at each time-step (Proposition 5.2).
>   These oscillations, however, do not appear detrimental in SD.
>
>   Together, this tradeoff characterizes the behaviour for large and small values of $\lambda$ and shows that the ideal range is tied to the magnitude of the scores.
>
>   We believe that smart ways to control the rigidity of the mixture dynamically are an interesting direction for future work, but a good approach may be tied to the particular semantics of a given diffusion process.
>   For example, we wonder if precisely chosen $\lambda$ at different points in time during diffusion can yield finer control over *how* images are composed.
>
> - **Larger logical formulas: Investigated in new SAT-Experiment (Appendix D)**
>   Thank you for this question! We decided to perform an additional toy experiment that emulates SAT:
>   We define models that sample from specific corners of a hypercube, such as those defined by propositional variables.
>   We then define logical formulas and verify whether the composition can successfully sample from the intended corners of the hypercube, excluding all others.
>   Similarly, we are able to sample with some success from an *exactly one* formula, which is written as $\left(p_1(x)\lor\ldots\lor p_k(x)\right)\land \bigwedge_{i,j\in [1,k], i\neq j} \left(\lnot_c p_i(x)\lor \lnot_c p_j(x)\right)$.
>   This formula heavily uses purely negated clauses, and even without FKC, we can sample with some success from this setup in 10 variables.
>   We will extend this experiment to a full ablation study and compare it to PoE with and without FKC.
> - **Sampling overhead: Negligible in practice, more efficient implementation for sampling without FKC.**
>     - **Computational Overhead with Oscillations**: This question is very interesting and not fully explored in our paper. Oscillating score mixtures likely reduce the quality of the resulting images, but it is not straightforward to quantify this in the sense of "lost sampling steps."
>    We believe that investigating this behavior is worth pursuing in future work, but it does not appear detrimental in SD.
>     - **Composition Construction Overhead**: In brief, not for small compositions, and not for large compositions if done well.
>     Composition time is linear in the number of components, yet many repeated operations can incur noticeable overhead in large formulas (like the 10-variable XOR experiment).
>     We do, however, believe that this is unlikely to be a practical issue.
>     Most computational overhead will be the unavoidable overhead from using multiple models.
>
> We again thank the reviewer for their positive comments and their interest in our work. Their inquiry to explore more complex compositions helped us improve our work.
> We hope we have addressed all their questions and look forward to further discussion.

---

### Official Review · Reviewer_ZuTu · 2025-10-31

**Soundness:** 3
**Presentation:** 2
**Contribution:** 3
**Rating:** 4
**Confidence:** 2

**Summary:**

This paper investigates compositional generation in diffusion models through the lens of fuzzy logic. By building on theoretical foundations in score-based generative models, existing compositional operators, and fuzzy set theory, the authors proposes Dombi operator that addresses limitations of prior approaches and demonstrates improved stability. Experimental results on small diffusion models, Stable Diffusion 1.4, and protein synthesis further support the effectiveness of the method.

**Strengths:**

- The paper is grounded in solid theoretical analysis, clearly identifying weaknesses in existing compositional operators for diffusion models.
- The proposed Dombi operator demonstrates strong compositional performance (e.g., Figure 3) and provides a flexible framework for combining multiple operators.

**Weaknesses:**

- The presentation of Sections 2-5 would benefit from improved clarity: several terms are introduced without sufficient explanation, and there are noticeable typos that hinder reading.
- Although the theoretical justification is strong, the empirical evaluation is relatively limited. Additional qualitative visualizations and ablation studies would help further substantiate the claims, particularly in the context of real-world diffusion models.
- Some empirical results for SD1.4 are referenced as being in the supplementary material (in Appendix C.2), but no supplementary was provided.

**Questions:**

- How does the Dombi operator affect inference-time efficiency when applied to large-scale diffusion models (e.g., Stable Diffusion)? Is the overhead negligible or significant?
    - How does the method extend to models with stronger inherent compositional abilities (e.g., SDXL, SD3)? Do the benefits persist or diminish?
- Were the images in Figure 3 generated with FKC correction? If so, how large is the performance gap between Dompi with and without FKC correction?

---

> ### Author Response · Authors · 2025-11-21
> **Review Response: Additional Experiment and further visualisations for Stable Diffusion**
>
> We thank the reviewer for their feedback and questions.
> The reviewer's requests inspired us to add a novel experimental setup to our manuscript and include further qualitative visualizations for our experiments with stable diffusion.
>
> ## Weaknesses
>
> - **Presentation: We rephrased the sections to improve clarity and incorporate feedback from all reviewers.**
>
> - **Ablation Studies: We set up an additional SAT experiment and report the results in our general answer above.**
> We set up an additional experiment where we analyze complex SAT-like compositions of diffusion models.
> We compose models that distribute particles to different corners of a hypercube, according to a provided formula (such as an $n$-variable XOR).
> In this setting, we have access to the ground truth of each diffusion process, allowing us to perform ablation studies with strong metrics.
> We provide preliminary results in the form of a table in our general response above.
> It can be seen that our composition method remains stable even for very complex compositions, such as an XOR over 10 variables, which is encoded using over a hundred operators.
> This is a qualitatively different result compared to PoE, which quickly grows unstable in compositions that heavily rely on negation.
> Additionally, we provide a qualitative Figure 8 in the Appendix of our manuscript, which visually compares Dombi Composition and PoE in this setting.
> We will include the results of a comprehensive ablation study in the camera-ready version of our paper.
> - **Qualitative Visualizations for SD experiment added to the appendix C.2: The theoretical tradeoff of $\lambda$ visualised in real-world diffusion models.**
>   Thank you for the request for more visualizations.
>   We now plotted the change rate of mixture coefficients as well as the log-likelihood ratios during conjunctive generation. This very nicely shows the tradeoff of the lambda parameter in stable diffusion:
>   High $\lambda$ gives good likelihood ratios, but variable mixtures, and vice versa.
>   This nicely illustrates the bounds from Section 5 in practice. We thank the reviewer for the suggestion.
>
> - **The supplementary material is now attached.** Due to file-size restrictions, only a subset of images could be provided.
>
> ## Questions
>
> - **Inference-Time Overhead: It is cheap, and there is no noticeable difference compared to other composition methods (PoE).**
>
>   During each timestep in the compositional generation, we need to:
>   - obtain the score of each of $k$ component models. This increases the computational complexity by a factor $k$, which is a fundamental and unavoidable overhead in model composition, but can be parallelized.
>   - Compute the density estimation increment, with the Itô Estimator, for each component. This requires only a few arithmetic operations and is negligible in terms of runtime.
>   - Compute the composition, e.g., according to Algorithm 1. This is just a softmax-weighted mixture for each operator.
>   In principle, this is also negligible, unless the number of operations in the composition is exceedingly large (exponential in the number of models).
>
> - **Larger-Scale SD Diffusion Experiments with FKC: Important direction for future Work.**
>   KC was not utilized in image generation for multiple reasons:
>     - SMC methods require larger batches of particles (images) to work, and with them, a lot of additional computing.
>     - Finding the best correction method and the best parameters requires careful tuning. Careless choice of resampling can result in either no impact on FKC or a complete collapse of the particle batch into the same image.
>     We recognize this as a shortcoming of this resampling approach and believe it is an important direction for future work to improve the stability of these correction methods.
>     However, they are not the focus of our approach.
>
> We again thank the reviewer for their time and effort in reviewing our work and look forward to further discussion!

---

### Official Review · Reviewer_9bJN · 2025-11-06

**Soundness:** 2
**Presentation:** 3
**Contribution:** 4
**Rating:** 4
**Confidence:** 4

**Summary:**

Combining or negating (conditional) diffusion models is an important problem for controllable generation, whether in text-to-image generation or designing molecules which (do not) have certain properties.   The paper highlights pitfalls with several existing heuristics for combining models via the product or negating models with quotients, and seeks to correct these by turning to fuzzy logic operations.    The proposed Dombi Operators satisfy DeMorgan's Laws and encompass several existing heuristics for combination or negation.     Finally, the authors derive a practical scheme for sampling from the combination of diffusion, combining techniques from several recent works.

The authors demonstrate gains from the proposed compositions on text-to-image generation and dual target drug design, along with several insightful examples to elucidate properties in Figure 2.

**Strengths:**

The paper is logically structured and well-executed in (i) demonstrating the pitfalls of existing heuristic for model combination, (ii) deriving Dombi Operators which satisfy DeMorgan's Laws as we would expect from logical operations (iii) deriving an SMC resampling scheme combining two recent methods (Ito Density Estimators and Feynman Kac Correctors), and (iv) demonstrating its efficacy.


Sec. 3.2 makes a *very* nice point about implicit temperature scaling of score addition!

I am very positive on the paper apart from a technical concern about the negation derivations (described below).   If valid, this should be mathematically salvageable but also may change experiment results.    Thus, I am giving a low initial score, but I am looking forward to discussion with the authors.

**Weaknesses:**

**Concerns re: Negation**
I have a concern regarding negation in Definition 4.1, especially moving from constant $c \perp x$ to $c(x)$.
- First, even for $c\perp x$, I get $\phi_c(x) = \frac{x}{x+c^2} \implies \phi_c^{-1}(y) = c^2 \frac{y}{1-y}$.
     - Now, $\phi_c^{-1}(1-\phi_c(p(z))) = \phi_c^{-1}( \frac{c^2}{p(z)+c^2}) = c^2\left( \frac{c^2}{p(z)+c^2} \right)\left( \frac{p(z)+c^2}{p(z)} \right) =\frac{ c^4 }{ p(z)}$
     - This does not match the stated $\neg_c p(z) = \frac{c^2}{p(z)}$ in Eq. 7

- What would it mean to take $\phi_{c(x)}(p(x))$?   This is using a different function $\phi_{\mathfrak{c}}(t)$ depending on the input $t \in [0, \infty]$ (since for given $x$, we are interested in $\mathfrak{c}=c(x)$ and $t = p(x)$ )
     - It is also unclear how to invert $\phi$ now
     - Not sure where the `order isomorphism' condition is used in Def A1, but again, now $\phi$ isn't a fixed function.

*Nevertheless, the conjunction and disjunction distribution operations in Eq 8-9 are correct for constant $c$.   Thus, only the negation operation is problematic*.   If the main goal for this definition of negation was to derive the contrast operator of Garipov et. al 2023, there should be other ways (see Observations below).

**Clarity**
The clarity of the paper is further lacking in several places.

*Definitions* (mostly fixed by including Def. A1 in main text)
- L215:  The role of $\phi_c$ has not been introduced.
    - $N(x) = 1-x$ corresponds to $\phi(x) = x$, and it probably does make sense to start with this $N(x)$.  You can then mention more general $N_\phi(x) = \phi^{-1}[ 1- \phi(x)]$ and reference App. Def A1  (*not referenced anywhere in main text!*)
    - This would also emphasize the role of $\phi$ *alone* to define negation, which then feeds through to conjunction and disjunction via DeMorgan's laws for a given $f$
- App Def. A1:  should have citation or proof that these satisfy DeMorgan's Laws (I confirmed that both hold for the given definition)
- L242:  $g(x)$ is not defined.


*Sec 5:  Bounds on Precision and Stability*
- Lines 274-275 and L276-277 should be reworded.   Only after reading the title of Corollary 5.1 "Idempotency and Distributivity Bias" did I understand what we we talking about.


*Algorithm*
- In Sec 6 (and possible Alg 1), it would be useful to re-emphasize the use of the Ito density estimator (Eq. 3) to track weights in the composite scores.

*Experiments*
- What $\lambda$ are used for operations in Figure 3?
- Please check the logic in Lines 396-398 and consider writing the result of each logical clause for legibility.  I get that $p_{\text{xor}} = \{ \textcolor{blue}{2,3}, \textcolor{orange}{0,1} \}$ so that $p_{\text{xor}} \wedge \neg p_3 = \{ \textcolor{blue}{3}, \textcolor{orange}{1} \}$ since $\{ \textcolor{blue}{2}, \textcolor{orange}{0} \} \in p_3$

**Questions:**

I am confused by the equation in Line 204, where the first and third logical statements are equivalent but the second is not.   I presume the authors wanted to obtain different "probability-distribution logic" for equivalent logical statements.



*Minor comments:*
- Prop. 6.1 should reference the Feynman-Kac / weighted PDE in Eq. 6.  Otherwise it is not clear where $g_t^{1,2}$ are coming from.
- why introduce the notion of energy?  I thought it might be cleaner to just use densities.
- In Alg. 1 line 4, the authors point out the transformation $\lambda \leftrightarrow -\lambda$ between conjunction and disjunction for fixed $\lambda$.   This is a useful observation worth emphasizing!
    - I suppose it's obvious from the score expressions in Lines 8-9, but less obvious from the distributions

**Observations:**

A distinct justification for the Dombi conjunction and disjunction in Eq 8-9 can be given using $\lambda$ (a.k.a. '$\alpha$' or power-) mixtures or quasi-arithmetic means of densities ([1], used e.g. for annealing paths in [2]).      For a similar generator $h(p(x)) = \frac{1}{\lambda}p(x)^{\lambda} - \frac{1}{\lambda}$, define
$$
\mu^{(\lambda)}_w(x; p,q) = h^{-1}[(1-w) h(p(x)) + w h(q(x))] / Z \\
= [ (1-w) p(x)^{\lambda} + w ~q(x)^{\lambda}]^{\frac{1}{\lambda}} / Z
$$
Now, for $w=1/2$, this term factors out and scales the mean by $(\frac{1}{2})^{\frac{1}{\lambda}}$ (similar to the factors in Corollary 5.1).   This $\lambda$ representation is closely related to the $\alpha$-divergence [1, Sec 4-5] (c.f. $\chi^2$-divergence in Eq. 10, see below)
This may be frivolous, or may be useful for the following reasons:

- $\lim_{\lambda \rightarrow 0} h(p(x)) = \log p(x)$, so $\mu^{(\lambda=0)}_w = p(x)^{1-w} q(x)^w / Z$, matching geometric averages or classifer-free guidance
    - for $w=1/2$, we get an `idempotent version' of the product ($\wedge$), i.e. $\mu^{(\lambda=0)}_{\frac{1}{2}} = p(x)^{\frac{1}{2}} q(x)^{\frac{1}{2}}/Z$
- Continuing with $w=\frac{1}{2}$ and emphasizing the $\lambda \rightarrow -\lambda$ relationship as in authors' Alg 1 Line 4, the product now fits nicely into the rest of the special cases described by the authors:
    - max:  $\lambda \rightarrow \infty$
    - mixture (MoE disjunction)  $\lambda = 1$
    - Dombi disjunction:  $\lambda > 0$
    - product (PoE conjunction) $\lambda \rightarrow 0$
    - Dombi conjunction:  $\lambda < 0$
    - harmonic mean:  $\lambda = -1$
    - min:  $\lambda \rightarrow -\infty$
- Perhaps it is not obvious to introduce $w \neq \frac{1}{2}$ from the fuzzy logic perspective (?), but quasi-arithmetic mixtures could expand the search space for effective model combinations.
- Logic is probably suitable for distributivity and chaining operations though.


*Garipov et. al 2023 Contrast?*

Given my concerns about negation as presented (and for fun), consider a different way to derive Garipov et. al 2023's contrast operator.   We want $p(x)$ 'and' $p(x)^2/q(x)$ (high p, low q, needs to be integrable)

$$
p(x) \wedge_{\lambda} \frac{p(x)^2}{q(x)}  = \frac{p(x)^3 / q(x)}{[ p(x)^{\lambda} + p(x)^{\lambda} \frac{p(x)^{\lambda}}{q(x)^{\lambda}}]^{\frac{1}{\lambda}}} = \frac{p(x)^3 / q(x)}{[  \frac{p(x)^{\lambda}}{q(x)^{\lambda}}\left( q(x)^{\lambda} + p(x)^{\lambda} \right)]^{\frac{1}{\lambda}}} =  \frac{p(x)^2}{\left( q(x)^{\lambda} + p(x)^{\lambda} \right)^{\frac{1}{\lambda}}}
$$
which matches for $\lambda = 1$.





[1] Amari 2007, Integration of Stochastic Models by Minimizing α-Divergence

[2] Masrani et. al 2021, q-paths: Generalizing Geometric Average Paths using Power Means

---

> ### Author Response · Authors · 2025-11-21
> **Review Response (1/3): Clarification of Negation**
>
> We are extremely grateful to the reviewer for the very detailed, insightful, and thoughtful review and incisive observations. We warmly appreciate the time and effort you put into providing many constructive suggestions, which we’ve gladly incorporated in the revised manuscript.
>
> A rough overview of our changes can be seen in our general response. In summary, we have addressed the concerns about negation as well as clarity, including the definition of $\phi_c$. We now also include in the manuscript a visualisation based on your perspicacious observation to demonstrate special cases as we vary $\lambda$.
>
> Below we address all your concerns and comments in detail.
>
> ## Negation:
> Many thanks for your meticulous perusal!
>
> **Apparent discrepancy due to a typo in Definition 4.1 (fixed in the revised manuscript)**
> You’re right - the apparent discrepancy between $\phi_c$ and $\lnot_c$ was caused due to an oversight for which we apologize.
>
> Specifically, the correct definition (that we in fact used throughout) should read as $\phi_c(x) = \frac{x}{x+c}$ **without squaring $c$**.
>
> This results in $\phi_c^{-1}(y) = c\frac{y}{1-c}$.
> Then, as intended, it holds that $\phi^{-1}_c(1-\phi_c(x)) = \phi^{-1}_c(\frac{c}{1-c}) = c\left(\frac{c}{x+c}\right)\left(\frac{x+c}{x}\right) = \frac{c^2}{x}$.
> This is fixed in the updated manuscript, along with removing energy $E$, as well as the (unused) helper function $g(x)$ from the definition entirely.
>
> **Role of order isomorphism**:
> Our operators are borrowed from Fuzzy Logic, where **membership**, i.e., the domain of propositional variables, is defined over the unit interval $[0,1]$.
> While there are other alternatives to match this definition of *densities* in $[0,\infty)$, we choose to match the domains with an order-isomorphism, effectively just "squashing" the extended reals into $[0,1]$.
>
> This approach aims to clarify the connection to the fuzzy logic literature (which describes and proves all the relevant properties) and reduce obfuscations. We now briefly mention this in the introduction when we introduce fuzzy logic, and in Section 4.
>
> **Why inversion is mathematically valid: Pointwise definition of operators**
> Thank you for the opportunity to clarify this, and apologies for the confusion due to abuse of notation. Please note that all our operators are defined strictly pointwise. In particular, we constrain $x$ to be the same for $p$ and $c$ when we define $\phi_{c(x)}(p(x))$. Thus, if $\phi$ is chosen differently for different $x$, this does not concern any property of the Dombi operators, and so long as $c$ is always the same function/distribution across a composition, the inverse is well-defined at each $x$.
>
> **Semantically chosen fixed-Point in Negation, Parametrization Invariance**:
> As the negation operator $n(x)$ is continuous and involute, it must have a fixed-point $x = n(x)$.
> Choosing any constant here is dangerous: scaling the parametrization of the distributions scales the densities, which can result in different "shapes" of negated distributions.
> The only meaningful neutral value with respect to fuzzy negation in our context is arguably some (unconditional) base distribution.
> This also brings the advantage of being affected by any change in parametrisation in an identical way to each operand of the negation.
>
> **Alternative perspective: Density ratio**
>
> An alternative to defining $\phi_{c(x)}$ to define operations directly over *density ratios*, implicitly accounting for the pointwise requirement.
>
> Specifically, the (now fixed) definition of $\phi_{c(x)}(p(x))$ is equivalent to $\phi_1({p(x)/c(x)})$.
> This resembles the versions of CFG and concept negation for energy-based models by Du and Liu, as acutely pointed out by Reviewer nPjQ.
>
> This perspective is nice for multiple reasons: early during the denoising process, all distributions converge to the same prior. Compositions in terms of density ratios result in a fixed point: if all operands are the same, all compositions are *precisely* idempotent, including the negation, if the reference $c(x)$ converges to the same prior.
>
> We have included the arguments presented here in the manuscript in Section 4 as well. We will dedicate a section in the appendix to elaborate on this further.
>
> We again thank for the detailed analysis and inquiries!

---

> ### Author Response · Authors · 2025-11-21
> **Review Response (2/3): Clarity and Connections to Power Means**
>
> ## Clarity
> Thank you very much for your suggestions toward enhancing clarity. We will act on all of these.
>
> **Definitions (fixed in the revised manuscript)**
> We have now introduced $\phi$ and clarified its role based on your suggestion. Also, as mentioned above, we have now been able to remove the (unused) helper function $g(x)$ from the definition entirely.
>
> **Proof of De Morgan's Duality**:
> We will add either a proof or a brief verbal argument to support this.
> Notably, our negation (for $c=1$) is just a reciprocal, flipping the sign of all exponents $\lambda$.
> If $c\neq1$, the double negation still cancels, leaving the definition of conjunction and disjunction unchanged.
>
> **Section 5. Bounds on Precision and Stability: renamed**.
> We have changed the section title to "Influence of $\lambda$ on distributivity and mixture stability".
> We will also rephrase the lines you pointed out to enhance the clarity of exposition.
>
> **Algorithm 1.: Ito Estimation now mentioned.**
> We will further emphasize the use of Ito density estimation in the text, as well as in the caption of Algorithm 1. This is a crucial aspect of our approach.
> Thank you for pointing this out.
>
>
> **Experiments: Details clarified.**
> Thank you for noticing these subtle details.
>  - The value of $\lambda$ is now clarified. We choose $\lambda=10$ for our experiments with SD and much smaller values $\lambda=5\cdot10^{-2}$ and $\lambda=5\cdot10^{-3}$.
>  The difference here is owed to the stark difference in score magnitudes of the models, and the greatly varying score magnitude in the small MNIST diffusion models, that use a variance exploding SDE.
>  - You are indeed correct, the resulting set should read $\{b3,y1\}$. We apologize for this mistake. It is now corrected in the manuscript, and we will provide the intermediary formula in text and images in the appendix in the camera-ready version.
>
>
>
> **Equation in Line 204: clearer explanation in the revised manuscript**:
> Thank you for the opportunity to reflect on this. The intention of this equation is to illustrate what we believe is a dangerous pitfall in equating PoE and reciprocals with logical operations: On first glance, it seems natural to avoid multiple distributions as $p(x) \land \lnot q(x) \land \lnot r(x)$.
> However, if $\land$ is a product and $\lnot$ is a reciprocal, this is equivalent to $p(x) \land \lnot (q(x)\land r(x))$.
> While the first formula intends to avoid $q$ and $r$ *individually*, the resulting composition will in fact avoid only their "intersection" $q(x)\land r(x)$. Consequently, carelessly using multiple negative guidance signals will lead to counterintuitive behavior. This is now better explained in the manuscript.Notably, in this example, the negative signals need to be mixed (MoE) first.
> In more complex compositions, this behavior is also disadvantageous from a combinatorial point of view: limited rewriting capabilities without DeMorgan's laws, Distributivity, and Idempotency prevent many tricks that enable a smaller---and more computationally efficient---formula, and with it, faster inference.
>
>
> ## Observations
>
>
> Many thanks for sharing these excellent observations!
>
> **Connection to power-mixtures**
> Thank you for the great pointers! We have included these references in the revised version of our manuscript, and they nicely complement our theory.
> The connection is correct, but care needs to be taken: The Dombi operators we propose are, in essence, (pseudo) power-norms over densities.
> As such, **Dombi composition is associative, DeMorgan-dual** and with bounded errors **idempotent and distributive** operators over **densities, log-likelihoods, and scores**.
> Power-means afford the same *score* calculus:
> For fixed $\lambda$, the scaling factor $\left(\frac{1}{2}\right)^{\frac{1}\lambda}$ between the power norm and the power mixture is constant, which leads to equivalent score operators. However, *power means are not associative over densities and log-likelihoods*.
>
> As you rightly observed, "Logic is probably suitable for distributivity and chaining operations though"---power norms afford us associativity, which is required to lift these operations from individual operations to a more general score calculus.
> So it seems that, while both justifications lead to essentially identical score operators, the properties of power-norms also hold for densities and log-densities (and give slightly nicer exponents).
> This is important as **we rely on associative likelihood operations**. In Algorithm 1, we aggregate likelihoods alongside the scores; here, the difference between power means and power norms can be observed.
>
> However, and thank you for pointing this out, as the score calculus is equivalent, the power means close the gap at $\lambda\to 0$ for our score calculus.

---

> ### Author Response · Authors · 2025-11-21
> **Review Response (3/3): New Figure and Further comments regarding the Reviewers' Observations**
>
> ## Weighted Composition
>
> Good point! Indeed, generalization is natural but the introduction of weighted composition is hard to motivate from the perspective of fuzzy logic. One of the most important aspects of our proposed operators is the chaining property, which allows for general compositions.
> Weighted, non-commutative operations are certainly useful, but we fear that they might dilute the focus of this work.
>
> **Connection to Garipov et al.**:
> Thanks for this elegant alternative derivation of the contrast operator from Garipov et al., bypassing the negation operator altogether!Indeed, we explicitly mention the connection to Garipov in the manuscript, and it is certainly beneficial to be able to analyse their contrast operator (via, or without, the negation operator) as well as other operators. However, our main motivation here for studying negation as an operator is due to several practical use cases that it enables in its own right.
>
> ## PoE with $\lambda \to 0$
>
> We discuss this point separately, to note two further details.
>  - The observation that PoE can be recovered (in score space, or for the power-mean definition of the operators), precisely at $\lambda \to 0$ illustrates a beautiful aspect of its behavior: when using a (inversely) multiplicative negation (which we argue is natural), there cannot be a dual operator for PoE.
>  - **We find the reviewer's overview of the unification of operators using lambda especially elegant and noteworthy!** It has inspired us to include a figure in our manuscript, which is now featured in the introduction, illustrating the logic as a whole and showing the connection to existing operators.
>
> We again express our gratitude to the reviewer for the comprehensive review and actionable suggestions that we believe have helped reinforce the many strengths of this work. We hope our response has addressed all your concerns, and this is reflected in a significantly upgraded score. We also welcome any further questions, suggestions, or clarifications.

---

> ### Comment · Reviewer_9bJN · 2025-11-21
> **reply to all**
>
> Thanks for the detailed responses, and I like the new figure!   My responses here are mostly for fun or comments about my own personal experience with the paper, and I intend to raise my score after clarification of the pointwise nature of negation.
>
> > We included a figure in the introduction, inspired by a comment from 9bJN, that illustrates our method and visually connects it to PoE, MoE, and the harmonic mean.
>
>  The figure took a while to parse...  I understand it as $q(x) = c (=.5)?$ being fixed on the left label, with shading indicating the composition as p(x) changes from 0 to 1.
> - If so, I think Operand $p(x)$ Likelihood might be better.    And identifying $c$ as the second operand.
> - Equations should be the same operation in the caption.   Maybe negation is fine as a comment but, of course, up to you.
>
> In any case, the behavior near λ =0 is very interesting.
>
> > much smaller values $\lambda=5\cdot10^{-2}$ and $\lambda=5\cdot10^{-3}$ (on MNIST?)
>
> It's interesting that the value of the operator can really blow up in these cases due to the $1/\lambda$ power (and is not in $[0,1]$!?).   Only when we aggregate the results into a normalized distribution do we get a result $\in [0,1]$, which is necessary for a "score" to be meaningful for the eventual FKC simulation.   In the end, the FKC scores and weights seem well behaved since the $\alpha^i$ weights are true mixture weights.
>
>
>
> > The intention of this equation is to illustrate what we believe is a dangerous pitfall in equating PoE and reciprocals with logical operations
>
> I am still confused by this equation in Sec 3.3, as the two equations do not match in *either* PoE/MoE operations or logical operations.   If binary logic (and Dombi?) would also require such care in order of conjunction operations (a reasonable baseline for 'expected behavior'), then I don't understand the goal of this argument.
>
> *Clarity of Appendix Derivations*
>
> - Not sure if I mentioned it first time around, but it would be nice to explicitly derive Definition 4.1 in App B (Prop A2 still requires substitution and only handles the score result)
>
> - The derivations in Lemma B1, moving from Eq 59 to Eq 60, are very non-trivial.
> Then the score of \nabla log p_t appears in Eq 62 without derivation.

---

> > ### Author Response · Authors · 2025-11-24
> > **Answer to further Comments and Clarification on small Lambda Effects and Equation in Section 3.3**
> >
> > We would like to reiterate our gratitude to the reviewer not only for their positive assessment but also for their continued interest in this work and their keen eye for detail.
> >
> > We are, of course, happy to discuss points that are for fun as well!
> >
> > **Thank you for the positive feedback regarding the new figure!**
> >
> > You correctly inferred that $q(x) = c$ but not necessarily $c=.5$.
> > the precise formula here is $p(x)\land c(x)$ with a flip of the y-axis presenting $\lnot_cp(x)\land c(x)$.
> > The figure looks the same for all $c$.
> > We will be more explicit in a revised version of the figure.
> >
> > **Composition with $\lambda \to_+ 0$ scales up densities, but scores are *always* a convex combination.**
> >
> > While we used the term "power-norm" during the discussion here, of course for $\lambda < 1$ the resulting expression does not pertain to an actual norm and diverges when applied over densities very close to 0.
> > However, for each $\lambda$ *constant* scaling means the mixture is normalizable, and for scores, $\alpha$ is determined by a softmax function, which means $\sum_i \alpha_i=1$.
> > For log-likelihoods, the additive bias is not as severe as it might seem on first glance:
> > For $|\lambda| \approx 10^{-3}$, we get biases of a few nats.
> > For the MNIST networks, the estimated log-likelihood is in the realm of hundreds of nats.
> > The bias introduced by the small $\lambda$ is not that large for these models. Furthermore, as all disjunctions are used "within" conjunctions (we encode formulas in CNF) the resulting densities are not scaled up drastically.
> > In general, using a conjunctive form of composition ensures **densities never scale up above the minimum**.
> >
> > **Equation in 3.3. shows modelling issues in compositions without DeMorgan**
> >
> > We're likely a bit verbose here in the interest of clarity. This equation intends to convey the following pitfall:
> >
> > In a direct logical interpretation, "Sample from A but not from B and not from C" would translate to $A\land\lnot B\land\lnot C$.
> > In other words, try to avoid settings where *either* of them is true.
> >
> > Applying DeMorgan's rule, we get $A\land (\lnot B \land\lnot C) = A\land \lnot( B \lor C)$ (in 3.3 the white and the green parts).
> > These formulas are equivalent in classical logic (and also in Dombi operators, which adhere to DeMorgan's Law *exactly*).
> >
> > PoE/MoE does **not** adhere to DeMorgans law, so $A\land (\lnot B \land\lnot C) \neq A\land \lnot( B \lor C)$ (the red and the green parts are not equal).
> > This is problematic if the right-hand side is the intended semantics, but the white part is what is modelled.
> > The left-hand-side is PoE-equivalent (**not** classical logic) to $A\land \lnot (B \land C)$, and will avoid the **intersection $B\cap C$ rather than the union $B\cup C$**.
> >
> > In score calculus this also becomes clear by considering that $A \land \lnot B \land \lnot C$ is PoE-equivalent to $s_A - s_B - s_c=s_A - (s_B + s_C)$, PoE equivalent to $A \land \lnot( B \land C)$. These two expressions should be distinct.
> > The equation in the paper shows the same calculation over densities instead of scores.
> >
> > In summary, this argument aims to convey **why** DeMorgan is necessary, as this Intuition mismatch cannot happen in Dombi Composition: We provide a detailed derivation that shows that in Dombi composition these two formulas are, in fact, the same (we show that DeMorgan holds here explicitly).
> >
> > \begin{align}
> > p_1\land \lnot p_2\land\lnot p_3
> > &=\left(p_1^{-\lambda} +\lnot p_2^{-\lambda} + \lnot p_3^{-\lambda} \right)^{-1/\lambda}\\\\
> > &= \left(p_1^{-\lambda} +\frac{c^{-2\lambda}}{p_2^{-\lambda}} + \frac{c^{-2\lambda}}{p_3^{-\lambda}} \right)^{-1/\lambda}\\\\
> > &= \left(p_1^{-\lambda} +c^{-2\lambda}\left(p_2^{\lambda}+p_3^{\lambda}\right) \right)^{-1/\lambda}\\\\
> > &= \left(p_1^{-\lambda} +\frac{c^{-2\lambda}}{\left(p_2^{\lambda}+p_3^{\lambda}\right)^{(1/\lambda)\cdot(-\lambda)}} \right)^{-1/\lambda}\\\\
> > &= \left(p_1^{-\lambda} +\frac{c^{-2\lambda}}{\left(p_2 \lor p_3\right)^{-\lambda}} \right)^{-1/\lambda}\\\\
> > &= \left(p_1^{-\lambda} +\lnot\left(p_2 \lor p_3\right)^{-\lambda}\right)^{-1/\lambda}\\\\
> > &=p_1 \land \lnot (p_2\lor p_3)
> > \end{align}
> >
> > **Clarity of Appendix Derivation: Additional derivations added.**
> >
> > We added the mentioned derivations to the manuscript now.
> > The raised point for Lemma B1 is fair: without the explicit derivation, the used identity can seem rather questionable.
> > We have now added additional steps to complete the exposition (now Eq. 77-86).
> >
> > We are thankful for the reviewer's keen interest, attention to detail, and continued engagement. We are happy to be held to a high standard and welcome any further suggestions for improvement.
> >
> > **Thank you so much!**

---

### Author Response · Authors · 2025-11-21
**General Response to the Reviewers and Overview of changes**

The combined feedback from the reviewers was incredibly insightful and helped us improve both the presentation and the theoretical contribution of our manuscript.
We are very excited about this work and wish to convey it to readers as clearly and effectively as possible.

Below is a brief list of the changes that have been implemented. These changes are preliminary, given the time constraints, and we will strive to further improve the manuscript with more time.

**For the ease of the reviewers, the sections with the most changes are temporarily presented in blue textcolor.**
## Implemented Changes
**Clarity: Relevant sections revised**. Reviewers noted typos and instances of unclear language. We revised the relevant sections, especially Definition 4.1, taking care to introduce and motivate each mathematical construct. We will further improve the writing.

**Negation in the manuscript and in related work: Clearer introduction and motivation in revision.**  We rephrased the introduction of $\phi_c$ as a mapping function, to more clearly argue for the motivation and correctness of our referenced negation. We revised the discussion of related work in concept negation. We plan to elaborate on this in the appendix. More details are provided in our response to 9bJN and nPjQ.

**Connections to Power-Means:** We now included a reference to work on ($\alpha$) power mixtures.
Suprisingly, these operators result in equivalent score calculus, but in terms of log-likelihoods and densities **only Dombi Composition forms a logic** as power means are non-associative.

**Additional Experiment with SAT: Investigates stability in complex compositions.** We include an experiment with complex composition formulas and provide preliminary results in this answer and the appendix. We will extend this to a thorough ablation study of our method in a controlled setting for the camera-ready.

**Additional figure in Introduction added.** We included a figure in the introduction, inspired by a comment from 9bJN, that illustrates our method and visually connects it to PoE, MoE, and the harmonic mean.

## Additional Experiment: SAT Diffusion

We propose a toy experiment where $k$ individual diffusion models act as propositional variables.
We analytically define each diffusion model to move particles towards Gaussians at the corners of a hypercube; each model targets **only one half of the hypercube**.
For example, one model targets only the corners on the right side, another targets the corners at the top, and so on.
A base model targets all corners.

We then define propositional formulas: XOR over $k$ variables, a majority formula, and an *exactly one* formula (OneHot).
We measure how many particles arrive at the correct modes and **how uniformly the modes are targeted**.
The compositions are highly complex, with *XOR* (encoded in 3SAT format) having over a hundred operators, and *exactly one* having many clauses that only contain negated distributions.
Our method still manages to reach mostly the correct modes in a uniform manner.
In contrast, PoE quickly grows unstable and exhibits shifts in the mode locations, especially when multiple negations are used.

The results below are preliminary; we will provide further details and more thorough results as soon as possible.

corr: fraction of particles at correct modes.
all: fraction of particles at any mode.
PPL: uniformity over particles in correct modes from 0-1. Higher is better.
| Formula        | Method | corr (k=2) | all (k=2) | PPL ↑ (k=2) | corr (k=5) | all (k=5) | PPL ↑ (k=5) | corr (k=10) | all (k=10) | PPL ↑ (k=10) |
|----------------|--------|--------------|-------------|-------------|--------------|--------------|--------------|----------------|---------------|---------------|
| **Maj_k**      | Dombi  | **1.00**     | **1.00**    | **1.00**    | **1.00**     | **1.00**     | **1.00**     | **1.00**        | **1.00**       | **0.98**       |
|                | Prod   | **1.00**     | **1.00**    | 0.82        | **1.00**     | **1.00**     | 0.09         | 0.00           | 0.00          | 0.00          |
| **XOR_k**      | Dombi  | 0.97         | **1.00**    | **1.00**    | **0.95**     | **1.00**     | **1.00**     | **0.91**        | **1.00**       | **0.98**       |
|                | Prod   | **1.00**     | 0.00        | **1.00**    | 0.00         | 0.00         | 0.00         | 0.00           | 0.00          | 0.00          |
| **OneHot_k**   | Dombi  | 0.97         | **1.00**    | **1.00**    | **0.59**     | **1.00**     | **1.00**     | **0.09**        | **1.00**       | **0.99**       |
|                | Prod   | **1.00**     | 0.00        | **1.00**    | 0.00         | 0.00         | 0.00         | 0.00           | 0.00          | 0.00          |

These results are included in the appendix, and an extended setup of this experiment will be provided for the camera-ready.
# We again thank all reviewers for their constructive feedback and look forward to further discussion!

---

### Author Response · Authors · 2025-12-01
**Statement for the Area Chair and Rebuttal Summary (1/2)**

We are grateful to the reviewers, (senior) area chairs,  program, and general chairs for their service. We particularly appreciate the extra efforts by the (newly assigned) AC due to the unforeseen circumstances this year.

# Statement to the AC
TL; DR: We received high-quality reviews in general and had a fruitful follow-up discussion.
While two reviewers did not get the opportunity to respond to our rebuttal, the other two acknowledged our efforts (with one of them raising their rating from 4 to 8, which resulted in an average updated rating of 6.5).
We first provide a brief general statement about the rebuttal period, followed by a summary of each individual discussion with the reviewers.
Below is a brief overview, followed by a more detailed summary:
- **9bJN, Rating 4 -> 8, Confidence 4 - detailed response to rebuttal**
  - Soundness of Negation: clarified to the reviewer and in the manuscript.
  - Clarity: Named issues fixed in the revision and appendix sections added.
  - Observations: Very helpful and incorporated in the manuscript.
  The reviewer was happy with the rebuttal and raised the score from 4->8.
- **ZuTu, Rating 4, Confidence 2 - did not yet respond**
  - Presentation: improved in manuscript revision.
  - Additional Ablation Studies and Figures: figures added, new ablation added in SAT-Experiment - will be further extended.
  - Supplementary Material containing images: now attached.
  - Inference time overhead: negligible.
  - FKC in image generation: no, in other experiments.
- **ndFP, Rating 8, Confidence 3 - did not yet respond**
  - Visual Clarity of Figure 2 (now Figure 3): will be improved based on the suggestion.
  - Introduction of mathematical concepts: now addressed.
  - Sensitivity of hyperparameter: clarified, covered by theory
- **nPjQ, Rating 6 -> 6, Confidence 3 - maintained score, no further discussion**
  - Du/Liu Style Negation: Now addressed in detail in the manuscript, theoretical justification added.
  - Du/Liu Style PoE: Discussed, same failure modes, used in experiments.

## Summary of the Rebuttal Period and Thanks to the Reviewers

We were fortunate to receive exceptional reviewers, who were excited about our work and helped us to improve the quality of our manuscript.
We want to express our special gratitude to reviewer **9bJN**, who held our manuscript to a very high standard, and provided very insightful observations.
We also thank **nPjQ**, whose recommendation for a more nuanced discussion of different negation variants also helped us to ground our manuscript.

## Summary of Individual Discussions and Points Raised By Reviewers (1/2)

### 9bJN, Rating 4 -> 8, Confidence 4

- **Concern about Validity of *referenced* Negation: technically sound and clarified in manuscript**.
  The reviewer raised two concerns about our negation derivations:
  - Mismatch between mapping function $\phi$ and $\neq$: This was caused by a typo in the definition of $\phi$ and has been addressed.
  - Concerns regarding a reference function $c(x)$ in negation: This was a misunderstanding caused by our notation $\phi_{c(x)}(p(x))$, which more exactly should be denoted as $\phi_{c}(p;x)$.
    We clarified this in the manuscript and explained that this can be seen as normalization.
- **Clarity: Improved in manuscript revision**.
  The reviewer noted a few areas for improvement in our manuscript, which we have addressed.
  These observations demonstrate the time and effort invested in the review, for which we thank the reviewer!
- **Observations by the reviewer - Connection to Power-Means: Very interesting and now mentioned in our manuscript**.
  The reviewer noted equivalent score operators can be derived from power means, which are power norms up to a constant factor (depending on $\lambda$).
  This observation, as noted by the reviewer, does not substitute our theory, as we rely on the properties of the fuzzy logic.
  The power-norms do not form a fuzzy logic, as they are not associative.
  However, the connection to power-means grounds our score operators with respect to concepts like alpha divergence.
- **Observations by the reviewer - Overview of $\lambda$ values: Very nice summary, now added to the manuscript as an introductory figure**.

#### **Summary**:
**The reviewer was satisfied with our clarification, and increased their score from 4 to 8.**
In further discussions, they noted we should add additional derivations to our appendix, which have since been added, and improve the self-containment of our manuscript.
Unfortunately, they did not have the opportunity to respond to our last message.

---

> ### Author Response · Authors · 2025-12-01
> **Statement for the Area Chair and Rebuttal Summary (2/2)**
>
> ## Summary of Individual Discussions and Points Raised By Reviewers (2/2)
> ### ZuTu, Rating 4, Confidence 2
> - **Weakness - Presentation and clarity in Section 2-5: Improved in Manuscript.**
>
> - **Weakness - Additional qualitative Visualizations and Ablation studies needed: Added additional experiment and multiple figures.**
>   The reviewer mentioned that the experimental evaluation would benefit from additional ablation studies and qualitative figures.
>   We addressed this concern by adding a figure (1) to the introduction, a figure (7) showing mixing behaviour in our stable diffusion experiment, illustrating the theoretical results in Section 5, and a new controlled experiment to investigate more complex compositions, including a figure (8) illustrating the setup and results in two dimensions.
> - **Weakness - Missing supplementary material containing generated images: now attached.**
> - **Questions - Inference-time overhead: negligible overhead on model forward-pass time.**
>   The reviewer asked if the Dombi Operators introduce significant overhead at inference time.
>   We clarified that the unavoidable inference time of the base models themselves is costly, and the composition is negligible in practice.
> - **Questions - FKC in Image generation?: Not used; effects studied in Protein Generation and will be studied in the new SAT experiment.**
> #### **Summary**:
> **The reviewer did not have a chance to react to our rebuttal.**
> We hope that we have adequately addressed their concerns.
>
> ### ndFP, Rating 8, Confidence 3
> - **Weakness - Visual Clarity of Qualitative Figure 2 (now Figure 3): We will improve the figure with more focus on Figure 2.b.**
> - **Weakness - Introduction of Mathematical Concepts: Addressed in Revision of the Manuscript.**
> - **Sensitivity of $\lambda$ and oscillations: Depends on the scale of the scores; small and large values do not drastically degrade the result.**
> The reviewer noted the large range of $\lambda$ values in our experiment.
> We noted that the tradeoff between large and small values is logical exactness versus mixing stability.
> Very small values of $\lambda$ behave like PoE, while very large values of $\lambda$ might lead to oscillation between the scores of different models.
> - **Larger Logical Formulas: Addressed in new Experiment**
> The reviewer asked about larger compositions.
> In our newly added SAT experiment, we addressed this with Compositions that rely on hundreds of operators and show that complex formulas like logical parity functions can be expressed and sampled from in a controlled setting.
> #### **Summary**:
> **The reviewer did not have a chance to react to our rebuttal.**
> This is very unfortunate, as we hoped for their feedback regarding our additional experiment.
> We hope we addressed their questions adequately and thank them for their enthusiasm.
>
> ### nPjQ, Rating 6 -> 6, Confidence 3
> - **Analysis of Du/Liu Style Negation (and Poe): Now addressed in the related Work section**
>   The Reviewer noted that our analysis of related work did not differentiate between two different forms of negation, which we now clarified and more precisely positioned ourselves with respect to.
>   The reviewer further noted the same for PoE.
>   **None of our identified failure modes in PoE disappear with the Du/Liu "density ratio" conjunctions**, the resulting operator (geometric mean for two operands) just shifts the problem, by uniformly decreasing temperature.
>
>   The reviewer mentioned that our theoretical justification for using this operator is still valuable. We collected arguments for our choice of operators and thank the reviewer for this valuable suggestion.
>   **As additional clarification: we note that our Stable Diffusion experiment actually uses the Conjunction mentioned by the Reviewer.**
> - **Question regarding Reparametrization Independence - Property is expressed by Dombi Composition**
>   We thank the reviewer for this recommendation and note that our operators indeed express this property.
>   As the reviewer noted, our operators are 1-homogeneous and pointwise. This guarantees reparemetrization independence according to [Bradley&Nakkiran25](https://arxiv.org/pdf/2502.04549).
>
> #### **Summary**:
> **The reviewer was satisfied with our clarification and maintained their score at 6.**
>
> # Conclusion:
> Thank you again for your service.  We're excited by the general positive response to this work, and the discussion period has helped reinforce the many strengths of this work.

---

### Meta-Review · Area_Chair_WQMg · 2026-01-04

**Summary:**

Reviewers were initially concerned regarding
* Theoretical issues
* Placement with respect to related work
* Clarity of presentation

**Reviewer Concerns:**

All concerns were addressed

**Reviewer Scores:**

9bJN, Rating 4 -> 8
ZuTu, Rating 4 -> 6
ndFP, Rating 8 -> 8
nPjQ, Rating 6 -> 6

Reviewers 9bJN and nPjQ were able to engage in discussion. I believe the concerns of ZuTu were successfully addressed. ndFP would have maintained their already positive score given the lack of major weaknesses.

---

### Decision · Program_Chairs · 2026-01-26

Accept (Poster)